# A co-assembly platform engaging macrophage scavenger receptor A for lysosome-targeting protein degradation

Qian Wang[1,2,3,4], Xingyue Yang[1,2,3,4], Ruixin Yuan[1,2,3], Ao Shen[1,2,3], Pushu Wang[1,2,3], Haoting Li[1,2,3], Jun Zhang[1,2,3], Chao Tian[3], Zhujun Jiang[1], Wenzhe Li[1] & Suwei Dong [1,2,3] ✉

Targeted degradation of proteins has emerged as a powerful method for modulating protein homeostasis. Identification of suitable degraders is essential for achieving effective protein degradation. Here, we present a non-covalent degrader construction strategy, based on a modular supramolecular co-assembly system consisting of two self-assembling peptide ligands that bind cell membrane receptors and the protein of interest simultaneously, resulting in targeted protein degradation. The developed lysosome-targeting co-assemblies (LYTACAs) can induce lysosomal degradation of extracellular protein IL-17A and membrane protein PD-L1 in several scavenger receptor A-expressing cell lines. The IL-17A-degrading co-assembly has been applied in an imiquimod-induced psoriasis mouse model, where it decreases IL-17A levels in the skin lesion and alleviates psoriasis-like inflammation. Extending to asialoglycoprotein receptor-related protein degradation, LYTACAs have demonstrated the versatility and potential in streamlining degraders for extracellular and membrane proteins.

Targeted protein degradation (TPD) is an emerging strategy for protein homeostasis modulation through directly degrading the protein of interest (POI) and eliminating its function[1]. TPD holds promise for targeting nonenzymatic proteins that were once considered 'undruggable' for inhibitor drugs[2]. TPD methodologies, such as proteolysis targeting chimeras (PROTACs)[3], autophagy targeting chimeras (AUTACs)[4], and lysosome targeting chimeras (LYTACs)[5], use dual targeting chimeric molecules to degrade proteins by hijacking ubiquitin-proteasome system (UPS), autophagy-lysosome system (ALS), or endosome−lysosome system (ELS). Notably, the majority of TPD methodologies hinge upon the use of bispecific chimeras to assemble a "ternary complex" with both the POI and receptor/E3 ligase[6]. In most cases, these covalently fused chimeras are composed of POI ligands and receptor/E3 ligase ligands, linked by a carefully chosen linker that requires an arduous optimization process, especially in the case of

PROTACs[7]. In the LYTACs strategies, researchers have emphasized the critical role of linker length in the glycan or peptide-based conjugates for receptor binding and the receptor-mediated degradation of targets[8,9]. Moreover, the activation of lysosome-targeting receptors, such as cation-independent mannose-6-phosphate receptor (CI-M6PR)[5] and asialoglycoprotein receptor (ASGPR)[10], often requires multivalent binding, demanding structurally complex and meticulously prepared ligands to effect an effective TPD process. Therefore, the development of a more flexible and general platform for incorporating multivalent ligands on demand would be desirable for the development of novel TPD strategies.

While PROTACs and AUTACs are limited to targeting proteins within the cytosolic domains, about 40% of encoded proteins are extracellular or membrane-bound, playing various roles in a multitude of diseases, such as neurodegenerative disorders, autoimmune

[1]State Key Laboratory of Natural and Biomimetic Drugs, Peking University, Beijing, China. [2]Chemical Biology Center, Peking University, Beijing, China. [3]Department of Chemical Biology, School of Pharmaceutical Sciences, Peking University, Beijing, China. [4]These authors contributed equally: Qian Wang, Xingyue Yang. ✉e-mail: dongsw@pku.edu.cn

diseases, and cancer[1]. To address this gap and target extracellular and membrane proteins for degradation, strategies utilizing the endosome–lysosome degradation pathway have been developed, including LYTACs that depend on the CI-M6PR[5] and ASGPR[6,10,11], KineTACs involving CXCR7[12], and the IFLD strategy that engages integrin[9]. Recently, the DENTAC strategy using scavenger receptors has been reported[13]. Despite these advances, suitable receptors for LYTACs and their biomedical applications remain to be further exploited.

In this study, we present a TPD approach that employs supramolecular co-assembly degraders derived from two peptide ligands with assembly-driven modification to selectively degrade interleukin-17A (IL-17A), a cytokine biomarker of autoimmune diseases with pro-inflammatory properties[14], engaging class A scavenger receptor (SR-A). This lysosome-targeting co-assembly (LYTACA) strategy is built using a designed SR-A peptide ligand (SRAL) and a previously reported IL-17A-binding peptide (17Abp) from our group[15]. These peptides are modified by aromatic motifs as a hydrophobic core that transforms them into co-assembled bispecific nanoparticles binding IL-17A and SR-A simultaneously. By exploiting the SR-A-mediated pathway for endocytosis and degradation of IL-17A by macrophages, we demonstrate the efficacy of the LYTACA approach. Furthermore, we show that this strategy can alleviate the psoriasis-like manifestation and skin inflammation induced by imiquimod (IMQ) in mouse models. Additionally, modifying LYATCA with a PD-L1 binder successfully results in the degradation of membrane PD-L1 on SR-A-expressing macrophages and cancer cells, showcasing its promise for application in cancer settings. Beyond the SR-A receptor, we further demonstrate that ASGPR could be targeted by LYTACA incorporating a mono-GalNAc binder, leading to successful PD-L1 degradation in HepG2 cells. These finding collectively highlights the versatility and potential of this method in targeted protein degradation.

## Results

### Design of a SR-A-mediated TPD strategy targeting IL-17A and discovery of a SR-A ligand peptide

In light of the pathological significance of IL-17A in autoimmune diseases and its therapeutic value as a drug target, an alternative approach to clear IL-17A may complement monoclonal antibody drugs that facing challenges in immunogenicity and the resulted anti-drug antibodies[16,17]. Prior to the recent DENTAC work by Zhu et al. [13], we noticed that SR-A is predominantly distributed on the surfaces of various immune cells, including macrophage, dendritic cells, and microglia[18,19]. Since SR-A can bind and eliminate negatively charged ligands via the lysosome-dependent pathways[20], they have been reported to mediate endocytosis of small molecule drugs, doxorubicin[21], and emtricitabine[20], conjugated with polymer-based SR-A binding ligands to target myeloid immune cells for therapeutics. We envisioned that a TPD strategy exploiting the lysosome-targeting functions of SR-A may lower the IL-17A levels in certain autoimmune diseases, such as psoriasis with symptomatic skin lesions where the macrophages infiltrate[22], and ultimately alleviate inflammation. To achieve this goal, a degrader incorporating ligands of IL-17A and SR-A is required.

To obtain a suitable and readily accessible SR-A ligand, we designed a 30-mer SR-A peptide ligand (SRAL), where glutamate and alanine residues were alternatively positioned (Supplementary Fig. 1a). Then we grafted fluorescein isothiocyanate (FITC) to the N-terminus of SRAL to investigate whether FITC-labeled SRAL (fi-SRAL) can enter macrophages and be transported to the lysosome. The results indicated that after treating RAW264.7 cells with fi-SRAL for specific time periods (1, 3, 6, 16 h), intracellular fluorescence proportionally increased and reached a maximum value at ~6 h (Supplementary Fig. 1b). Notably, colocalization of the fi-SRAL fluorescence signal with the lysosome was also observed (Supplementary Fig. 1c). These

findings suggest that SRAL may serve as a promising candidate ligand for engaging SR-A in TPD.

### A mixture of SR-A- and IL-17A-binding peptides enables cellular uptake of exogenous IL-17A by macrophages

Next, we constructed a chimeric molecule by linking the SRAL peptide and a stapled IL-17A-binding peptide (17 Abp), which we previously developed and demonstrated its superior binding affinity and proteolytic stability[15] (Fig. 1a). These two motifs were covalently linked using 6-aminocaproic acid (Acp) as a spacer, and further labeled with FITC at its N-terminus to afford the covalent SRAL-17Abp chimera (fi-SRAL-17Abp). Subsequently, we incubated RAW264.7 cells with exogenous IL-17A and the fi-SRAL-17Abp chimera for 6 h, with control groups treated with fi-SRAL, fi-17Abp, and a mixture of fi-SRAL and fi-17Abp peptides. The internalized IL-17A was stained with an anti-IL-17A fluorescent antibody and detected by flow cytometry.

Surprisingly, the covalent fi-SRAL-17Abp chimera did not increase IL-17A uptake in cells. In contrast, the mixture of fi-SRAL and fi-17Abp peptides showed a remarkable increase in intracellular IL-17A signal compared to the control group, whereas no obvious enhancement was observed in cells treated with fi-SRAL or fi-17Abp alone (Fig. 1b). To validate this result, we embarked on examining the effects of this peptide mixture at different concentrations and for different incubation times. We observed an upsurge in the uptake efficiency of IL-17A as the peptide concentration increased (Fig. 1c). Moreover, the process was time-dependent, with the intracellular IL-17A fluorescence signal reaching the maximum at 6 h and then decreased to the initial level at 12 h (Fig. 1d). Contrastingly, the IL-17A level in the culture medium exhibited no discernible alterations, as confirmed by western blot analysis (Supplementary Fig. 2), indicating an overdose of the exogenously added protein. Simultaneously, we monitored changes in SR-A abundance through flow cytometry. The results revealed SR-A recovery at 48 h following a decline at 24 h (Supplementary Fig. 3), supporting the idea of SR-A recycling during the degradation process. Consequently, we propose that the decline in intracellular IL-17A signal is likely associated with the consumption and reduced concentration of the mixed peptides during the latter stage of the time-course experiment, rather than a depletion of SR-A. Notably, in clinical settings, the reported concentration of IL-17A in psoriatic lesions is 0.01 ng/ml[23], significantly lower than the concentration (0.5 µg/ml) we used in our in vitro cellular assay. Therefore, the efficiency of LYTACA would be sufficient to achieve the degradation of IL-17A at levels relevant to those observed in patients. Collectively, these results clearly demonstrated that the non-covalent mixture of fi-SRAL and fi-17Abp can mediate the internalization and degradation of IL-17A in RAW264.7 cells.

### SR-A involves in the endocytosis of IL-17A induced by peptide mixture of fi-SRAL and fi-17Abp

To investigate the mechanism underlying the endocytosis of IL-17A induced by the peptide mixture of fi-SRAL and fi-17Abp, we performed competitive inhibition assays from both IL-17A and SR-A binding perspectives. Since peptide 17Abp competes for the same binding domain of IL-17A with IL-17RA[15,23], we employed IL-17RA as a competitor. Cells were treated with varying concentrations of IL-17RA in the presence of fi-SRAL and fi-17Abp mixture, along with IL-17A. The intracellular IL-17A signal exhibited a concentration-dependent attenuation contingent on IL-17RA levels (Fig. 1e). Moreover, an additional competition assay was performed using an SR-A competitive inhibitor, polyinosinic acid (poly I), and a non-inhibitor control, polycytidylic acid (poly C)[20]. RAW264.7 cells were incubated with varying concentrations of Poly I or Poly C in the presence of exogenous IL-17A and the peptide mixture. As expected, poly I co-treatment resulted in concentration-dependent inhibition, while poly C exerted no discernible impact on the uptake efficiency (Fig. 1f). To further validate SR-A engagement in IL-17A

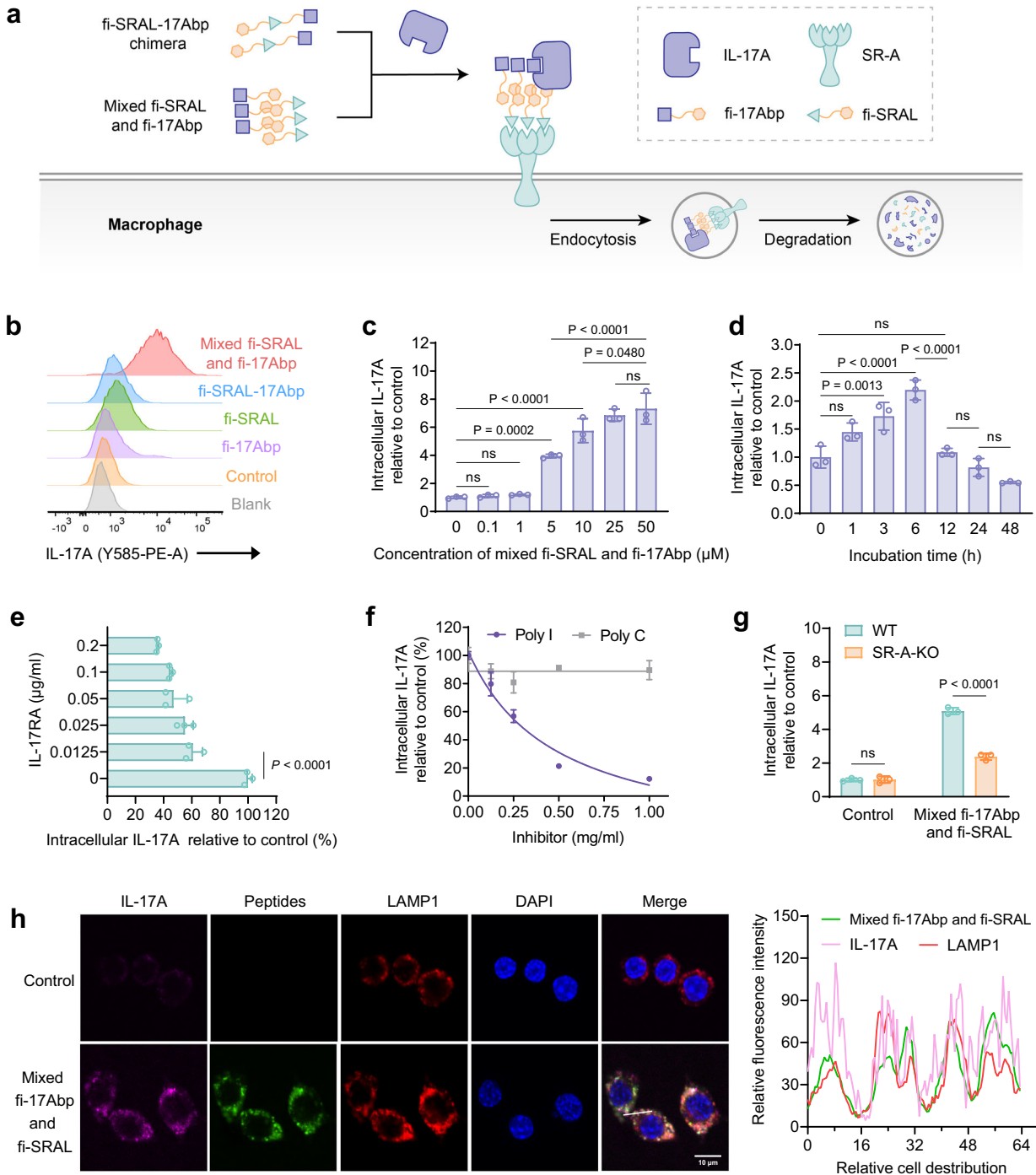

**Fig. 1 | SR-A-engaged lysosome-targeted endocytosis of IL-17A in RAW264.7 cell lines induced by non-covalent ligands. a** Schematic illustration of SR-A-mediated lysosomal degradation of IL-17A in macrophage using the fi-SRAL-17Abp chimera or the mixed fi-SRAL and fi-17Abp. **b** Representative flow cytometry results showing a shift in mean fluorescence intensity (MFI) in RAW264.7 cells treated for 6 h with 2 μg/ml IL-17A and 50 μM mixed fi-SRAL and fi-17Abp compared to treatment with fi-SRAL-17Abp, fi-17Abp or fi-SRAL alone. Control: treated with 2 μg/ml IL-17A, Blank: treated without IL-17A. **c** Dose escalation experiment for IL-17A uptake in RAW264.7 cells incubated with 0.5 μg/ml RED-tris-NTA-labeled IL-17A and mixed fi-SRAL and fi-17Abp at the indicated concentrations for 6 h. **d** Time-course experiment for IL-17A uptake in RAW264.7 cells incubated with 0.5 μg/ml IL-17A and 50 μM mixed fi-SRAL and fi-17Abp for the indicated time. **e** IL-17A binding competition experiment for IL-17A uptake in RAW264.7 cells treated with 0.5 μg/ml IL-17A, 50 μM mixed fi-SRAL and fi-17Abp, IL-17RA at the indicated concentrations for 6 h. **f** SR-A inhibition assay in RAW 264.7 cells treated for 6 h with 0.5 μg/ml IL-17A, 50 μM mixed fi-SRAL and fi-17Abp, along with poly I (SR-A inhibitor) or poly C (control) at the indicated concentrations. **g** Uptake of IL-17A in wildtype RAW264.7 cells (WT) or SR-A knockout RAW264.7 cells (SR-A-KO) incubated with 0.5 μg/ml IL-17A and 50 μM mixed fi-SRAL and fi-17Abp for 6 h. **h** Confocal microscopy images depict RAW264.7 cells treated for 6 h with 0.5 μg/ml RED-tris-NTA-labeled IL-17A and 50 μM mixed fi-SRAL and fi-17Abp. Scale bar, 10 μm. The intensity profiles along the white line are plotted in the right panel. For **c**–**g**, MFI was determined by flow cytometry and relative to the control (cells treated with 0.5 μg/ml IL-17A). Data are presented as the mean ± SD, and error bars represent the standard deviation of biological replicates ($n = 3$). $P$ values were determined by one-way ANOVA with Tukey's multiple comparisons test (**c**–**e**) or two-way ANOVA with Sidak's multiple comparisons test (**g**). Source data are provided as a Source Data file.

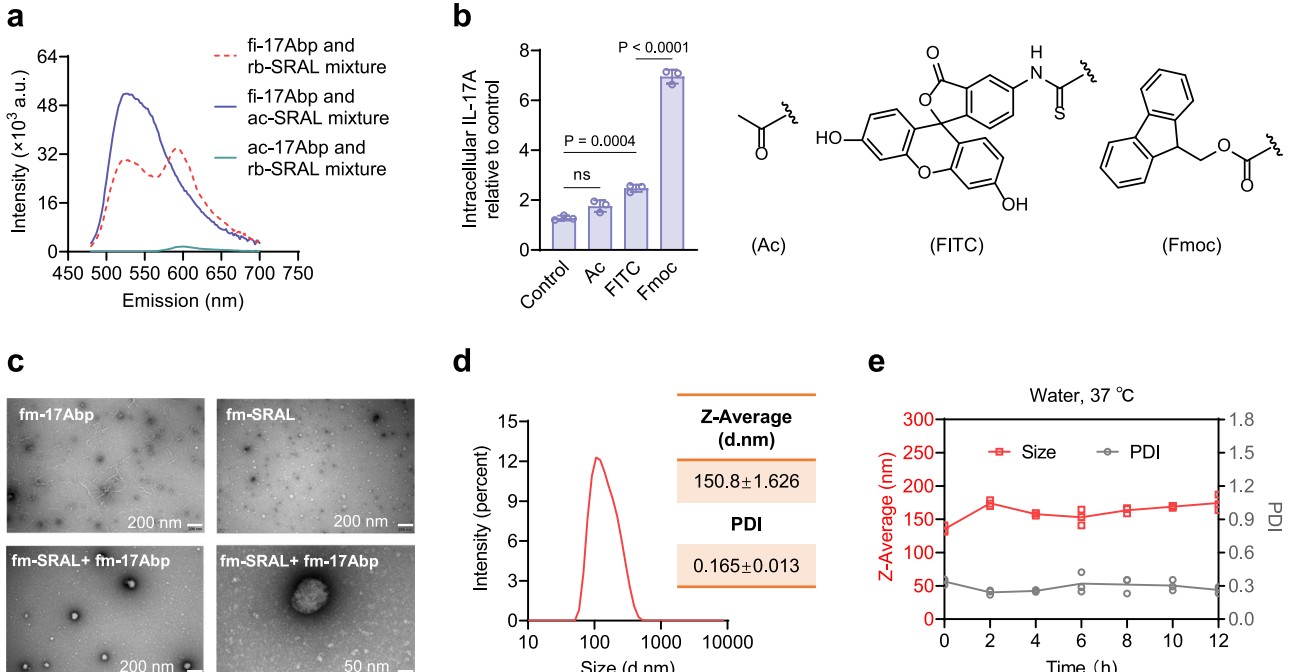

**Fig. 2 | The molecular mechanism of mixed 17 Abp and SRAL for facilitating the IL-17A endocytosis. a** Fluorescent emission spectra of the mixed fi-17Abp and rb-SRAL (red dotted curve), the mixed fi-17Abp and ac-SRAL (purple curve), and the mixed ac-17Abp and rb-SRAL (green curve). The peptides were mixed in a 1:1 molar ratio, and the excitation wavelength was 450 nm. ac-SRAL: N-acetylated SRAL. ac-17Abp: N-acetylated 17 Abp. **b** The influence of N-terminal labeling group including acetal group (Ac), FITC, and Fmoc to the IL-17A endocytosis. Left: uptake of IL-17A in RAW264.7 cells treated for 6 h with 0.5 μg/ml IL-17A, 50 μM mixed ac-SRAL and ac-17Abp/ mixed fi-SRAL and fi-17Abp/ mixed fm-SRAL and fm-17Abp. MFI was determined by flow cytometry and relative to the control (cells treated with 0.5 μg/ml IL-17A). Right: chemical structure of N-terminal modification group acetyl, FITC, and Fmoc. **c** Representative TEM images for fm-17Abp, fm-SRAL, and the mixed fm-17Abp and fm-SRAL (two images with different scale bars) at a concentration of 400 μM in H₂O. Scale bar, 200 nm or 50 nm. **d** Hydrodynamic size and polydispersity index (PDI) of the mixed fm-17Abp and fm-SRAL (n = 3). **e** Time-dependent changes in the hydrodynamic size (red) and PDI (gray) of the co-assembled supramolecular complex formed by fm-17Abp and fm-SRAL at 37 °C. For **b** and **e**, data are presented as the mean ± SD, and error bars represent the standard deviation of biological replicates (n = 3). *P* values were determined by one-way ANOVA with Tukey's multiple comparisons test. Source data are provided as a Source Data file.

endocytosis, we established an SR-A receptor knock-out cell line (SR-A-KO) with RAW264.7 using the CRISPR technology (Supplementary Fig. 4). The endocytosed IL-17A signal in SR-A-KO cells was significantly lower than in WT RAW264.7 cells, providing conclusive evidence for the involvement of SR-A in IL-17A uptake (Fig. 1g).

We also tested the effect of low temperature (4 °C) incubation or treatment with different endocytic pathway inhibitors on IL-17A uptake. Incubation at 4 °C[24] or treatment with NaN₃[25] significantly reduced intracellular fluorescence signals of IL-17A (Supplementary Fig. 5a) suggesting that the uptake of IL-17A by macrophages is an energy-dependent process. Compared to the positive control group, we observed significant decreases in cellular fluorescence in the groups treated with the clathrin-endocytosis inhibitor chlorpromazine (CPZ) and the lipid rafting pathway inhibitor methyl-β-cyclodextrin (Me-β-CD) (Supplementary Fig. 5b). These evidence also support the involvement of SR-A, which had been reported to undergo internalization from the plasma membrane via clathrin-dependent endocytosis[26] or a lipid raft-dependent mechanism[27]. Furthermore, treatment with the pinocytosis inhibitor 5-(N-ethyl-N-isopropyl)-amiloride (EIPA) also decreased IL-17A uptake, suggesting that the pinocytosis pathway may also be involved. Confocal experiments showed that the peptide mixture-treated group resulted in strong intracellular fluorescent signal of RED-tris-NTA-labeled IL-17A, which colocalized with lysosome indicated by LAMP1 (Fig. 1h). Collectively, these experimental data suggest that the fi-SRAL and fi-17Abp peptide mixture efficiently facilitates the trafficking of exogenous IL-17A to lysosomes for degradation in macrophage, where SR-A plays an essential role in the endocytosis process.

## Peptides co-assembly enables targeted IL-17A degradation by macrophages

It was noticed that the endocytosis of IL-17A increased with an extended pre-incubation of the peptides fi-SRAL and fi-17Abp (Supplementary Fig. 6a), leading to the suspicion of potential interactions between these two components that may have formed a supramolecular complex. To confirm this hypothesis, fluorescence resonance energy transfer (FRET) experiments were conducted on the mixture of fi-17Abp and rhodamine B (RB) labeled SRAL (rb-SRAL) in an aqueous solution (Fig. 2a). Excitated by a 450 nm wavelength fluorescence, the mixture of fi-17Abp and non-fluorescent ac-SRAL displayed a strong fluorescence peak at 530 nm (characteristic emission of FITC), while the mixture of ac-17Abp and rb-SRAL showed a weak fluorescence peak at 595 nm (characteristic emission of RB). For the mixture of fi-17Abp and rb-SRAL, the fluorescence at 530 nm was weakened while that at 595 nm was increased sharply, indicating FRET occurrence and the potential formation of co-assembled supramolecular complex. The FRET signal of two peptides was also be observed in RAW264.7 cells (Supplementary Fig. 6b), further corroborating the conclusion. Intriguingly, we found that the replacement of the N-terminal FITC groups in those two peptides with 9-fluorenylmethyloxycarbonyl (Fmoc) further enhanced the IL-17A uptake efficiency, while acetyl (Ac) groups resulted in insignificant change compared to the control group (Fig. 2b). This result suggested that the hydrophobic aromatic ring at the N-terminus plays a significant role in mediating the interaction between the two peptide ligands.

To evaluate whether supramolecular structures were formed, we conducted transmission electron microscopy (TEM, Fig. 2c) and dynamic light scattering (DLS, Fig. 2d) analysis on the Fmoc-labeled

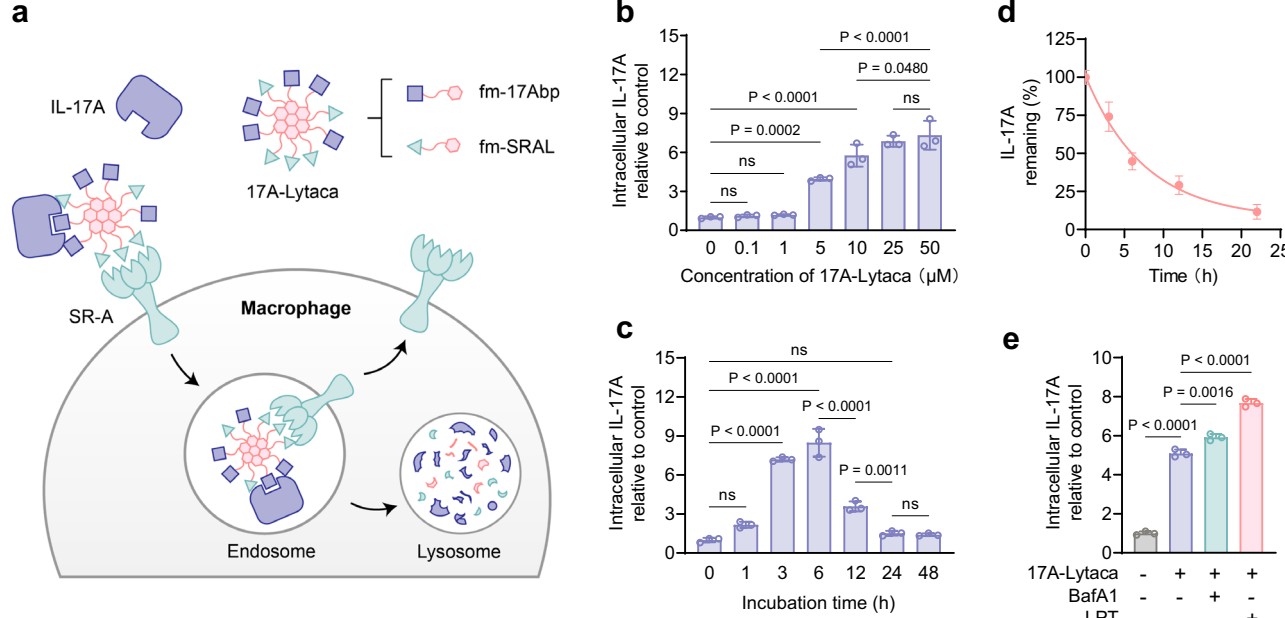

**Fig. 3 | 17A-Lytaca induced the lysosomal degradation of IL-17A. a** Schematic of targeting IL-17A for lysosomal degradation via SR-A-mediated macrophage endocytosis induced by 17A-Lytaca. **b** Dose escalation experiment for IL-17A uptake in RAW264.7 cells incubated with 0.5 μg/ml IL-17A and 0.1, 1, 5, 10, 25 or 50 μM fm-17Abp-Lytaca for 6 h. **c** Time-course experiment for IL-17A uptake in RAW264.7 cells incubated with 0.5 μg/ml IL-17A and 50 μM 17A-Lytaca for 0, 1, 3, 6, 12, 24 and 48 h. **d** Temporal IL-17A degradation experiment. The intracellular IL-17A signal was assessed at the indicated time after a 6 h treatment with 0.5 μg/ml IL-17A and 50 μM 17A-Lytaca. **e** Lysosome inhibition assay. The intracellular IL-17A signal was assessed after treatment with 0.5 μg/ml IL-17A and 50 μM 17A-Lytaca for 6 h in the presence or absence of 100 nM BafA1 or 0.1 mg/ml leupeptin (LPT). For **b**–**e**, MFI was determined by flow cytometry and relative to the control (cells treated with 0.5 μg/ml IL-17A). Data are presented as the mean ± SD, and error bars represent the standard deviation of biological replicates (*n* = 3). *P* values were determined by one-way ANOVA with Tukey's multiple comparisons test. Source data are provided as a Source Data file.

17Abp and SRAL mixture (fm-17Abp and fm-SRAL). The data showed that the peptide mixture formed monodisperse and spherical nanoparticles with a hydrodynamic diameter of 150.8 ± 1.626 nm. In contrast, fm-SRAL alone formed smaller-sized nanoparticles than the mixed peptides, and fm-17Abp alone generated short fibers. To gain more insight into the nature of the peptide co-assembly, we conducted a surfactant competitive assay[28] using sodium dodecylbenzene sulfonate (SDBS) as a competitor. The TEM images showed that the mixture of fm-SRAL and fm-17Abp in SDBS-containing buffer could not form the same spherical nanoparticles as it did in water (Supplementary Fig. 6c). This disruption of co-assembly is likely caused by interactions between the aromatic moiety of SDBS and the Fmoc groups, supporting the aromatic-driven self-assembly process. Assessed by DLS[29], the critical aggregation concentration for the co-assembly was determined as 11.75 μM (Supplementary Fig. 6d), and this supramolecular complex maintained the size and monodispersity over a 12-h period at both 37 and 4 °C. (Fig. 2e, Supplementary Fig. 6e). Furthermore, we investigated the stoichiometry of the co-assembled complex in varying molar ratios. After a 12-h incubation, the samples underwent centrifugation to remove the aggregates and subsequent ultrafiltration to eliminate the residual peptides (Supplementary Fig. 6f). High-performance liquid chromatography (HPLC) analysis of the resulting materials revealed a stoichiometry of ~1:7 for fm-SRAL and fm-17Abp (Supplementary Fig. 6g). The impact of 17A-Lytaca stoichiometry, with varying molar ratios, on IL-17A intake efficiency was also evaluated, confirming that the optimal ratio for preparing the complex is 1:1 of fm-SRAL and fm-17Abp (Supplementary Fig. 6h). Collectively, these results support the notion that mixed fm-SRAL and fm-17Abp peptides can co-assemble into nanoparticles, which may be the active form that interacts with both IL-17A and SR-A, thus facilitating the cell uptake of extracellular IL-17A.

As Fmoc-modified peptides demonstrated the highest efficiency in promoting IL-17A endocytosis (Fig. 2b), we selected this pair, named

it 17A-Lytaca (17A-Lysosome-targeting co-assembly), for further studies. Derived from aromatic and hydrophobic tail-modified peptide binders of IL-17A and SR-A, 17A-Lytaca shows promising potential in inducing lysosomal degradation of IL-17A by macrophages (Fig. 3a). This is evidenced by a dose escalation experiment in which an upsurge in the uptake efficiency of IL-17A was observed as the peptide concentration increased (Fig. 3b) and a time-dependent increase in intracellular IL-17A signal, peaking at 6 h before returning to the basal levels by 48 h (Fig. 3c). Notably, the treatment of 17A-Lytaca activated the SR-A-mediated signaling response, evidenced by an increase in extracellular regulated protein kinase (ERK) phosphorylation (Supplementary Fig. 7), further supporting the involvement of SR-A in our system. Conversely, the Fmoc-modified covalent peptide chimera with varying linker lengths failed to promote the uptake of IL-17A (Supplementary Fig. 8). This observation may be explained by the intrinsic property of SR-A, which requires multivalent binding provided by a branched polyanionic ligand, as seen in the reported polymer-based structure[20]. Hence, the single-chain peptide in the covalent chimera might not be sufficient to bind SR-A effectively.

Moreover, after a 6-h treatment with IL-17A and 17A-Lytaca, followed by replenishing the culture medium without these components, a time-dependent reduction of the endocytosed IL-17A signal was observed, indicating a complete degradation of the endocytosed IL-17A in macrophage cells over ~22 h (Fig. 3d). To further confirm the involvement of lysosomes in the 17A-Lytaca-promoted IL-17A degradation, cells were treated with the lysosome inhibitor Leupeptin (LPT) and Bafilomycin A1 (BafA1). Both inhibitor-treated groups exhibited significantly higher IL-17A accumulation in cells, indicating suppression of lysosomal degradation (Fig. 3e). Additionally, different endocytosis efficiency of IL-17A induced by 17A-Lytaca was observed in various cell lines (Supplementary Fig. 9a), including RAW264.7, U-118 MG, DC 2.4, and HaCaT, which are known to express the different level of SR-A. The western blot result showed that the RAW264.7 cells

expressed the highest amount of SR-A among these cell lines (Supplementary Fig. 9b, c), consistent with the IL-17A uptake result. Combined with the cell viability assay conducted with varying concentrations of 17A-Lytaca (Supplementary Fig. 10), these data highlight the safety and potential of our LYTACA strategy for selective cellular targeting.

### LYTACAs clear IL-17A and alleviates psoriasis symptoms in vivo
Given the therapeutic roles of IL-17A in psoriasis[30–33] and the pathological features of macrophage infiltration around the epidermal-dermal interface in psoriasis patients[34,35], we conducted in vivo experiments to evaluate the efficacy of our IL-17A-clearing LYTACAs for treating psoriasis. We envisioned that by utilizing the abundant SR-A-expressing macrophages in psoriatic skin lesions, the LYTACA treatment may down-regulate the pathogenetic IL-17A level and alleviate the inflammation. Accordingly, we employed an IMQ-induced psoriasis-like mouse model[36], where BALB/c mice were treated with IMQ cream on their shaved back skin for seven consecutive days, followed by intraperitoneal injection of 17A-Lytaca, control-Lytaca (co-assembly prepared from peptide fm-nbp that does not bind to IL-17A, Supplementary Fig. 11), methotrexate (MTX), fm-17Abp alone, fm-SRAL alone, or saline as control (Fig. 4a).

As anticipated, the results showed that 17A-Lytaca treatment successfully reduced the severity of psoriasis manifestation compared to the positive group. This was evidenced by factors such as body weight loss (Fig. 4b), splenomegaly (Fig. 4c), both clinical and pathological characteristics (Fig. 4d), acanthosis (Fig. 4e), and disease severity, including a reduction in skin thickness, erythema, and scaling (Fig. 4f, g, Supplementary Fig. 12). Interestingly, treatment with control-Lytaca, fm-17Abp or fm-SRAL alone showed no significant difference compared to the saline group, highlighting the superiority of our co-assembly strategy over inhibitor treatment. In addition, the 17A-Lytaca treatment demonstrated better performance in slowing down weight loss (Fig. 4b) and reducing the PASI score (Fig. 4f) than the MTX group, underscoring the superior biocompatibility and safety of our co-assembly system. Notably, the 17A-Lytaca treatment did not show any obvious toxicity to the organs of mice (Supplementary Fig. 13).

To further validate the IL-17A-targeting capability of 17A-Lytaca in vivo, we measured the IL-17A levels in mouse dorsal skin. Impressively, the 17A-Lytaca treatment resulted in a significant reduction of IL-17A in the skin lesion (Fig. 4h). In contrast, the control-Lytaca, fm-SRAL- or fm-17Abp-treated group exhibited no obvious effects on changing IL-17A levels compared to the IMQ model group. Remarkably, treatment of 17A-Lytaca markedly decreased the mRNA expression of IL-17A and its downstream psoriasis-related inflammatory cytokines in the skin lesion, including IL-6, CCL-20, and IL-22, compared with the positive group. Conversely, the control-Lytaca, fm-17Abp, and fm-SRAL alone showed no difference from the saline group (Fig. 4i).

The clearance kinetics of the peptide components within 17A-Lytaca were also investigated by fluorescently labeling fm-SRAL with Cy3 and fm-17Abp with Cy5. Both peptides' fluorescent signals were detectable by in vivo imaging system (IVIS) within the initial 8 h. Subsequently, fm-SRAL disappeared, while fm-17Abp persisted until 24 h in healthy mice (Supplementary Fig. 14). In psoriasis mice, both peptides remained detectable until 48 h (Supplementary Fig. 15). The results also indicated that the co-polymer could reach and persist in the skin lesion and spleen in IMQ-induced psoriasis mice, while the relevant organs of healthy mice showed no peptide signal (Fig. 4j–m). Collectively, these results demonstrate that our co-assembly strategy effectively degrades IL-17A both in vitro and in vivo and displays potent anti-inflammatory activity, successfully ameliorating psoriasis symptoms in the IMQ-induced mouse model.

### LYTACAs target PD-L1 for degradation
After demonstrating the LYTACA system's ability to degrade extracellular protein in vitro and in vivo, we sought to investigate whether this strategy could be extended to membrane-bound proteins. As a proof of concept, we targeted programmed death-ligand 1 (PD-L1), as it is expressed on both tumor cells and tumor-associated macrophages (TAMs), playing crucial roles in suppressing tumor-specific T cell immunity[37]. Extensive studies have shown that blocking PD-L1 not only enhances T-cell recognition of tumor cells but also remodels macrophages from an anti-inflammatory state to a pro-inflammatory one, enhancing the immune response against tumors[38]. Therefore, degrading PD-L1 in TAMs may improve cancer immunotherapies. To this end, we carried on the study by first generating a fm-PD-L1-Lytaca system, which combines a Fmoc-Acp-modified PD-L1-binding peptide[39] (fm-TPP) with fm-SRAL (Fig. 5a). This mixture was subsequently applied to incubate RAW264.7 cells, along with the control groups using fm-TPP alone or PBS buffer. After 8 h, cell-surface expression of PD-L1 was analyzed by flow cytometry. The fm-PD-L1-Lytaca reduced the level of PD-L1 compared to the control group, while fm-TPP alone had no degradation effect (Supplementary Fig. 16a). Further examination of PD-L1 levels in cells treated with fm-PD-L1-Lytaca for different periods revealed that the degradation effect began between 4 and 8 h and lasts up to 12 h (Supplementary Fig. 16b). Concentration-dependent responses were observed in PD-L1 degradation in RAW264.7 cells treated with fm-PD-L1-Lytaca, peaking at 100 μM. In contrast, neither fm-TPP nor fm-SRAL alone showed any discernible impact on degradation at a concentration of 50 μM (Fig. 5b). Importantly, the absence of PD-L1 degradation in SR-A-KO cells (Fig. 5c) conclusively demonstrates the involvement of SR-A in this TPD process.

Given that scavenger receptors (SRs) are highly expressed in cancer cells[13,40,41], we selected A549 and HepG2 cells to ascertain the capability of our LYTACA in degrading PD-L1 on tumor cells. Western blot results revealed successful degradation in both cell lines (Fig. 5d, Supplementary Fig. 16c), with HepG2 exhibiting superior performance, achieving over 50% degradation. To enhance degradation efficiency, we sought to optimize the structures of the assembling motifs. Considering the influential roles of the hydrophobic N-termini in increasing endocytosis (Fig. 2b), we prepared an improved PD-L1-Lytaca by replacing Fmoc with Nap-FF (abbreviated as nf), a structure with robust assembly-driving capability[42–44] owing to enhanced π-π stacking (Fig. 5e). TEM showed that the nf-modified PD-L1-Lytaca formed spherical nanoparticles (Fig. 5f), while nf-TPP and nf-SRAL alone formed fibers (Supplementary Fig. 17). Remarkably, compared with fm-PD-L1-Lytaca, nf-PD-L1-Lytaca exhibited significantly enhanced degradation efficiency in macrophages (Fig. 5g) and cancer cells (Supplementary Fig. 18a, b). The reduction of PD-L1 induced by nf-modified Lytaca proved to be both concentration and treatment time-dependent, reaching a peak degradation exceeding 70% in RAW264.7 cells (Fig. 5h, i, Supplementary Fig. 18c). Notably, during this TPD process, the SR-A level exhibited an initial decrease followed by recovery over time (Fig. 5i), suggesting that Lytaca did not result in the depletion of SR-A. Collectively, these findings underscore the potential of LYTACAs to induce the degradation of cell membrane proteins in macrophages and tumor cells through the involvement of SR-A.

To showcase the versatility of our LYTACA strategy across different receptors, we investigated the asialoglycoprotein receptor (ASGPR), primarily expressed on hepatocytes, which recognizes glycoproteins bearing N-acetyl galactosamine (GalNAc) or galactose (Gal)[45–47]. Previous studies indicate that interactions with a degrader containing multivalent ligands, such as tri-GalNAc, can enable ASGPR to facilitate lysosome-targeting protein degradation[6,10,11]. In this study, we employed a newly designed PD-

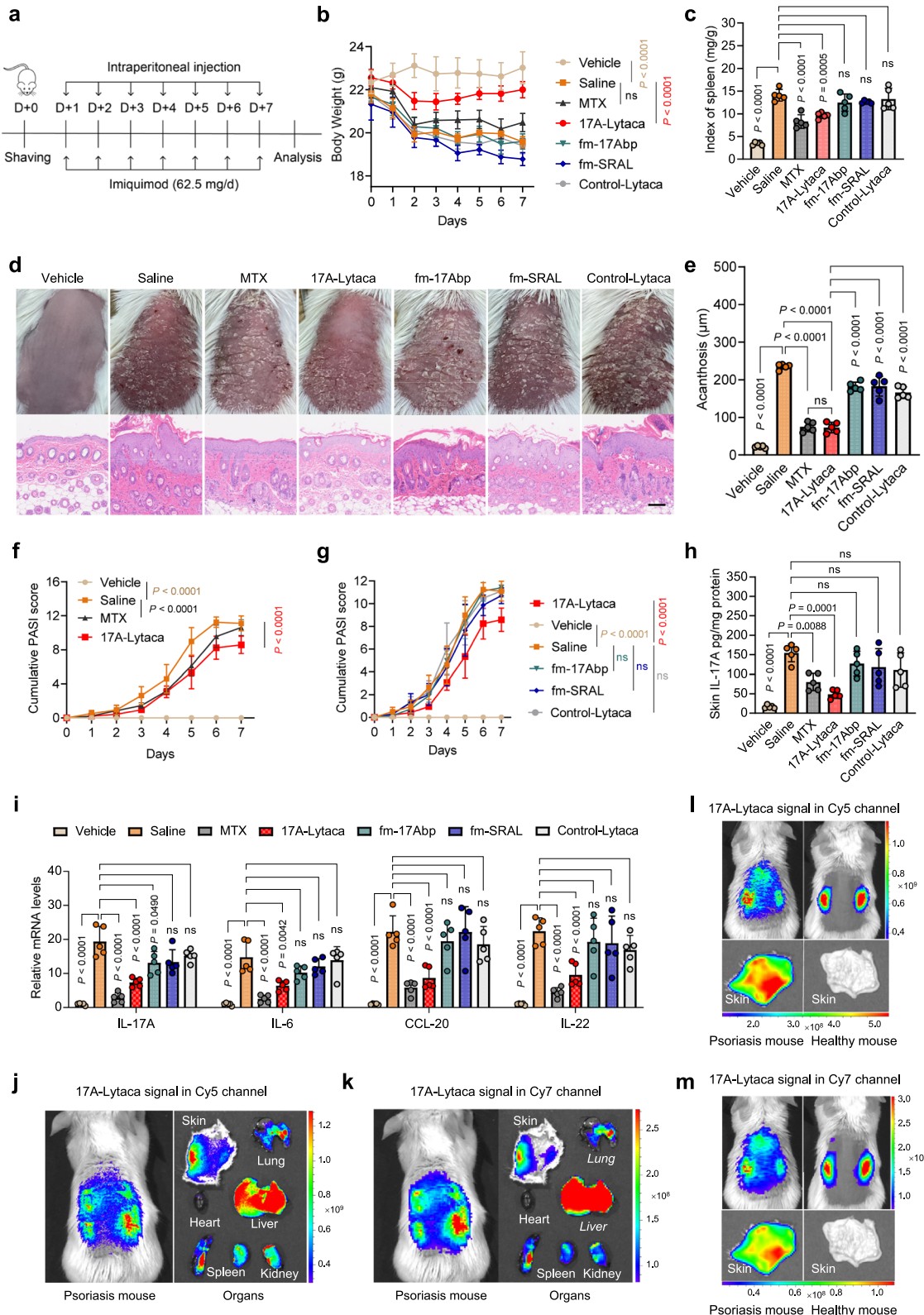

L1-Lytaca equipped with ASGPR-targeting mono-GalNAc and PD-L1-targeting TPP warheads (Fig. 5j). The successful reduction of PD-L1 levels in HepG2 cells (Fig. 5k) not only demonstrates that the LYTACA strategy extends beyond the SR-A receptor but also underscores a distinct advantage: the effect of a readily accessible mono-valent structure can be amplified to achieve effective inter-action with receptors that typically demand complex multivalent ligands.

## Discussion

In conclusion, we have introduced an innovative assembly strategy for constructing protein degraders, offering a modular platform to develop effective structural entities capable of inducing TPD for both extracellular and membrane proteins (Fig. 6). Leveraging the assembly-driven aromatic cores, the resulting supramolecular co-assembly LYTACAs demonstrated the ability to degrade IL-17A and PD-L1 in a time, concentration, lysosome, and receptor SR-A-dependent

**Fig. 4 | LYTACAs attenuate skin inflammation in psoriasis mouse model via IL-17A clearance. a** Experimental timeline. For 7 consecutive days, Saline, fm-17Abp (15 mg/kg/day), fm-SRAL (22 mg/kg/day), 17A-Lytaca (fm-17Abp and fm-SRAL mixture was given 15 and 22 mg/kg/day after preincubated for 12 h), control-Lytaca (fm-nbp and fm-SRAL mixture was given 12 and 22 mg/kg/day after preincubated for 12 h) or MTX (1 mg/kg/twice day) were intraperitoneally administered into BALB/c mice after the topical application of 5% IMQ cream. **b** Daily body weight changes. Data are presented as the mean ± SEM, and error bars represent the standard deviation of biological replicates (*n* = 5). **c** Splenomegaly degree. **d** Representative clinical manifestations and H&E staining of the back skin on day 7, scale bar: 100 μm. **e** Analysis of acanthosis change on day 7. **f, g** PASI score reflecting the severity of skin erythema, scaling, and thickening for 7 days. **h** IL-17A protein level in skin lesion on day 7. **i** Relative mRNA expression of IL-17A, IL-6, CCL-20 and IL-22 in skin lesion. **j, k** Representative in vivo and organ fluorescence images (*n* = 3) of psoriasis mice at 6 h after intraperitoneal treatment with Cy-labeled 17A-Lytaca, comprising Cy5 labeled fm-17Abp (5 mg/kg) and Cy7 labeled fm-SRAL (7.3 mg/kg). **l, m** Representative in vivo and skin fluorescence images of psoriasis mice or healthy mice at 6 h after intraperitoneal treatment with Cy-labeled 17A-Lytaca, comprising Cy5 labeled fm-17Abp (5 mg/kg) and Cy7 labeled fm-SRAL (7.3 mg/kg). For **c** and **e–i**, data are presented as the mean ± SD, and error bars represent the standard deviation of biological replicates (*n* = 5). *P* values were determined by one-way ANOVA with Tukey's multiple comparisons test (**c, e, h, i**) or two-way ANOVA with Sidak's multiple comparisons test (**b, f, g**). Source data are provided as a Source Data file.

manner. With successful PD-L1 degradation observed across various cell lines, including ASGPR-expressing HepG2, we highlight LYTACA's excellent modularity in incorporating different binder motifs targeting the POIs and the receptors. In addition, in comparison to covalent chimeras that demand a strenuous optimization process, this co-assembly degradation system is relatively straightforward to prepare using diverse peptide binders, thereby curtailing resource and time expenditure. Furthermore, LYTACA facilitates multivalent binding for POIs and receptors, which can potentially augment the degradation efficacy while streamlining the design and synthesis of degraders for multivalent receptors such as SR-A and ASGPR. Notably, during the preparation of this manuscript, the Li group reported a split-and-mix PROTAC (SM-PROTAC) nanoplatform for the degradation of intracellular protein[48], which validates the versatility of co-assembling complex and could be complimentary to our approach.

One of the major challenges in developing PROTACs is the lack of cell or tissue selectivity, which can lead to potential systemic toxicity[2,49]. Given that SR-A receptors are widely distributed on the surface of immune cells, our LYTACA strategy may offer advantages in targeting cells and tissues selectively and may be beneficial for reducing on-target toxicity. In an illustrative example using a mouse model of psoriasis, 17A-Lytaca demonstrated sustained presence in macrophage-infiltrated skin lesions for up to 48 h. It proved highly effective in alleviating psoriasis manifestation by eliminating the pathogenic factor IL-17A. Potentially, the ability to selectively target specific tissues may provide benefits for an optimal therapeutic window.

While our study highlights the potential of LYTACA in treating immune disorders, further investigations are necessary to fine-tune the LYTACA system for specific applications and to assess its safety and effectiveness in both preclinical and clinical settings. Notably, LYTACAs might face the challenge of being endocytosed by cells before effectively capturing the POI, a challenge common to all current methods in the targeted degradation of extracellular proteins. One way to circumvent this issue might involve optimizing the degrader to achieve faster and more efficient binding to the targeted protein. This optimization could be accomplished through various approaches, and our co-assembly strategy offers one relatively convenient option. Continued efforts to identify more suited degraders could significantly broaden the scope of targeted protein degradation and pave the way for discovering new treatments for various diseases.

## Methods
### Ethics statement
Our research complies with all relevant ethical regulations. Animal protocols were approved by the subcommittee of the Experimental Animal Welfare Ethics Committee of the Biomedical Ethics Committee of Peking University (approval number: LA2022226). 8–10-week-old female BALB/c mice which are widely used in the IMQ-induced Psoriasis model were used for the in vivo experiments. All mice were housed in an SPF animal facility (light/dark cycle: 12 h/12 h, temperature: 20–26 °C, humidity: 40–70%) with ad libitum access to food and water. After the end of the experiment, the mice were immediately euthanized.

### General peptide synthesis procedures
Automated solid-phase peptide synthesis was performed using Fmoc-Rink Amide-MBHA Resin with a loading level of 0.355 mmol/g, purchased from Gill Biochemicals. $^{\alpha}N$-Fmoc-protected amino acid compounds for peptide solid phase synthesis were sourced from various brands including Gill Biochem, Bomagex, CS bio, Biotek, NovaBiochem, and Energizer: Fmoc-Ala-OH, Fmoc-Asn(Trt)-OH, Fmoc-Asp(OtBu)-OH, Fmoc-Cys(Trt)-OH, Fmoc-Glu(OtBu)-OH, Fmoc-Glu(OAll)-OH, Fmoc-Gly-OH, Fmoc-His(Trt)-OH, Fmoc-Ile-OH, Fmoc-Leu-OH, Fmoc-Lys(Dde)-OH, Fmoc-Lys(Boc)-OH, Fmoc-Phe-OH, Fmoc-Pro-OH, Fmoc-Thr(tBu)-OH, Fmoc-Trp(Boc)-OH, Fmoc-Val-OH. The CS Bio peptide synthesizer (CX136XT) was used for automated peptide synthesis. Peptides were synthesized using DMF as a solvent, with deblocking in piperidine/DMF (20/80, v/v) for 5 min (×2) and coupling for 20 min using excess amino acids (4 equiv.) and HATU/HOBt (1:1, 4 equiv.) as coupling reagents. The coupling cycle was repeated as necessary for amino acids after sterically hindered residues like Pro, Ile, Thr, and Val.

N-terminal Modification. Following solid-phase synthesis, the peptide-loaded resin was transferred to the peptide synthesis tube with DCM for N-terminal modification. For acetyl-protected, a mixed solution of 5 ml of DMF/acetic anhydride/DIEA (8/1/1, v/v/v) was added and the reaction liquid was discharged after shaking on a shaker for 5 min. For FITC or rhodamine B-labeled, 10 equiv. of DIEA and 1.2 equiv. of FITC/Rhodamine B were added, the peptide synthesis tube was wrapped with tinfoil to avoid light, and the reaction liquid was discharged after shaking overnight on a shaker. The remaining resin in the synthesis tube was washed three times with DCM-DMF-DCM and dried on a vacuum pump for 15 min.

Resin cleavage and global deprotection were performed in a cocktail solution of TFA/H$_2$O/TIPS (95:2.5:2.5, v/v/v) for 2 h. The resin was removed by filtration, and the filtrate was concentrated under a nitrogen atmosphere. The resulting residue was washed with cold diethyl ether to give a white solid, which was then dissolved in a mixture of acetonitrile and water containing 5% acetic acid. The resulting solution was ready for HPLC purification after filtration.

### HPLC−MS analysis and preparative HPLC
All HPLC separations involved a mobile phase of 0.05% (v/v) TFA in water (solvent A), and 0.04% (v/v) TFA in acetonitrile (solvent B).

Analytical HPLC−MS chromatographic separations were performed using a Waters Alliance e2695 Separations Module, an SQ Detector, and a Waters 2489 UV/Visible (UV/Vis) Detector equipped with an Agilent C18 column (5.0 μm, 4.6 × 150 mm) at a flow rate of 0.4 ml/min, or with a Welch-XB C4 column (3.0 μm, 3.0 × 150 mm) at a flow rate of 0.3 ml/min. The wavelengths of the UV detector were set to 210 and 220 nm.

Preparative HPLC separations were performed using a Hanbon Science & Technology NP7005C solvent delivery system and a Hanbon

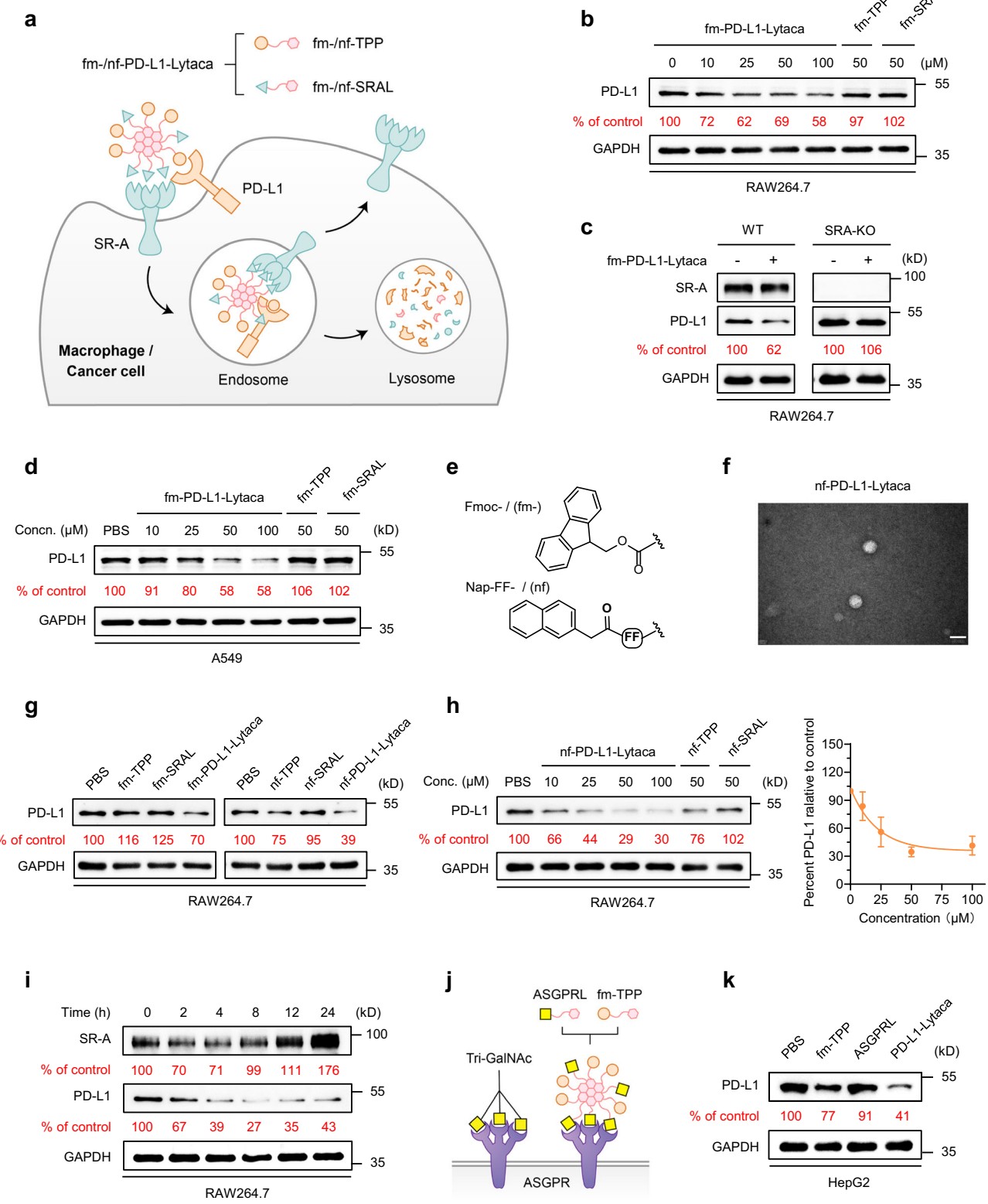

Science & Technology NU3010C UV detector equipped with an Exsil Pure 300 C18 column (10.0 μm, 20 × 250 mm) at a flow rate of 15 ml/min, or a Welch-XB C4 column (5 μm, 10 × 250 mm) at a flow rate of 4 ml/min. The wavelengths of UV-detector were set to 210 and 220 nm.

## Chemical analysis instrumentation

[1]H NMR spectra were recorded at 400/600 MHz at ambient temperature with CDCl3 or CD3OD (Cambridge Isotope Laboratories, Inc.) as

the solvent unless otherwise stated. [13]C NMR spectra were recorded at 151 MHz at ambient temperature with CDCl3 or CD3OD as the solvent unless otherwise stated. Chemical shifts are reported in parts per million relatives to CDCl3 ([1]H, δ 7.26; [13]C, δ 77.0), CD3OD ([1]H, δ 3.31; [13]C, δ 49.0). Data for [1]H NMR are reported as follows: chemical shift, integration, multiplicity (s = singlet, d = doublet, t = triplet, q = quartet, m = multiplet), and coupling constants ($J$ Hz). All [13]C NMR spectra were recorded with complete proton decoupling. High-resolution mass

**Fig. 5 | LYTACAs enable degradation of the membrane protein PD-L1.**
**a** Schematic illustration of SR-A-engaged lysosomal degradation of PD-L1 in macrophages or cancer cells using the fm-/nf-PD-L1-Lytaca, which was prepared from a solution with a 1:1 molar ratio of the fm-/nf-TPP and fm-/nf-SRAL peptides. **b** Dose escalation experiment for PD-L1 degradation in RAW264.7 cells treated with 10, 25, 50 and 100 μM fm-PD-L1-Lytaca or 50 μM fm-TPP/fm-SRAL for 12 h. **c** Western blot of PD-L1 in wild type (WT) or (SR-A-KO) RAW264.7 cells treated with or without 50 μM fm-PD-L1-Lytaca for 12 h. **d** Dose escalation experiment for PD-L1 degradation in A549 cells treated with 10, 25, 50, and 100 μM fm-PD-L1-Lytaca or 50 μM fm-TPP/fm-SRAL for 12 h. **e** Chemical structure of N-terminal modification group Fmoc and Nap-FF. **f** Representative TEM image of nf-PD-L1-Lytaca at a concentration of

400 μM in H$_2$O ($n = 3$). Scale bar, 50 nm. **g** Western blot of PD-L1 in RAW264.7 cells treated with 50 μM fm-/nf-PD-L1-Lytaca, fm-/nf-TPP or fm-/nf- SRAL for 12 h. **h** Dose escalation experiment for PD-L1 degradation in RAW264.7 cells treated with 10, 25, 50, and 100 μM nf-PD-L1-Lytaca or 50 μM nf-TPP/nf-SRAL for 12 h. **i** Western blot of PD-L1 and SR-A in RAW264.7 cells treated with 50 μM nf-PD-L1-Lytaca for 2, 4, 8, 12, 24 h. **j** Schematic illustration of ASGPR binding with Tri-GalNAc or fm-PD-L1-Lytaca consisting of Fmoc-modified TPP ligands and mono-GalNAc ligands. **k** Western blot of PD-L1 in HepG2 cells treated with 50 μM fm-TPP, ASGPRL or PD-L1-Lytaca (fm-TPP + ASGPRL) for 12 h. For **b–d**, **g–i**, **k**, densitometry was used to calculate protein levels, and data ($n = 3$ biologically independent experiments) were normalized to the control. Source data are provided as a Source Data file.

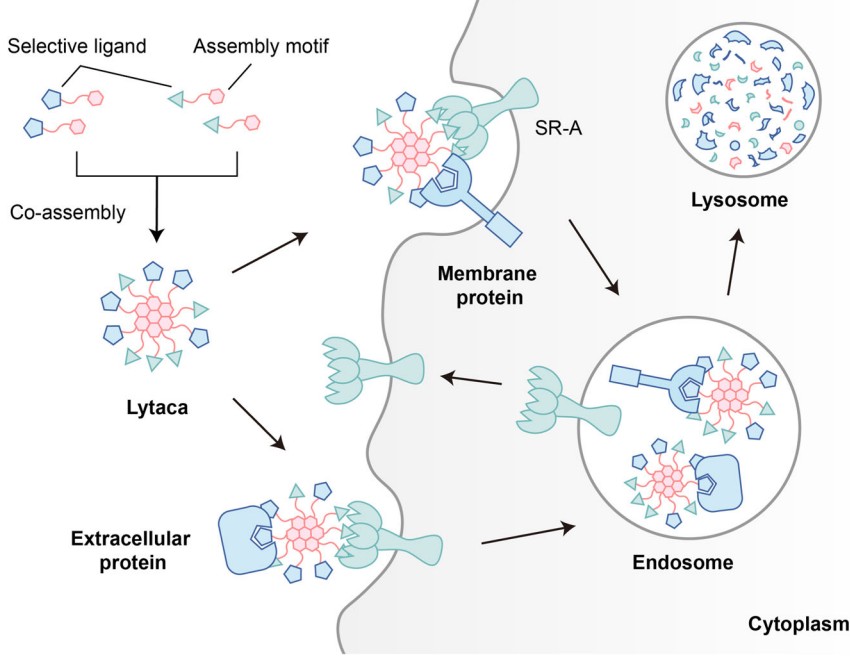

**Fig. 6 | LYsosome-TArgeting Co-Assembly (LYTACA) strategy for protein degradation of both extracellular and membrane proteins.** LYTACA is based on a modular supramolecular co-assembly system consisting of two selective ligands with an assembly motif. The constructed non-covalent degrader, Lytaca, can bind cell membrane receptors and the protein of interest simultaneously, resulting in internalization and subsequent lysosome degradation.

spectra were obtained in the Chemical Instrumentation Center, Peking University Health Center using a Waters Q-TOF mass spectrometer (Xevo G2Q-TQF).

### Cell culture
RAW264.7(1101MOU-PUMC000146) and HepG2(1101HUM-PUMC 000035) cells were obtained from the Cell Resource Center, Peking Union Medical College (PCRC). U-118MG cells were purchased from iCell (Cat# iCell-h217). HaCaT cells were purchased from MeisenCTCC (Cat#CTCC-OO2-OO12). DC2.4 and A549 cells were generous gifts from Prof. Peng Chen's Lab and Prof. Houhua Li's Lab, respectively, at Peking University. All cells were cultured in complete Dulbecco's modified Eagle medium (DMEM, Gibco) supplemented with 10% fetal bovine serum (PAN) and 1% penicillin–streptomycin (M&C GENE) at 37 °C and 5% CO$_2$. The human astroglioma U-118 MG cells (iCell) were cultured in DMEM with a low concentration of NaHCO$_3$ (iCell) supplemented with 10% fetal bovine serum (PAN) and 1% penicillin–streptomycin (M&C GENE) at 37 °C and 5% CO$_2$. All cell lines regularly tested negative for mycoplasma contamination.

### Generation of knockout-cell line
SR-A-deficient cell line was generated using the CRISPR-Cas9 system in the following steps.

*Preparing DNA templates for sgRNA*: DNA templates for sgRNA were amplified by PCR with gRNA scaffold and primers (Tsingke Biotechnology) containing a T7 promoter and crRNA sequence (TGAACGTGCGTCAAATTTCA). The sequences of gRNA scaffold and primers are listed in Supplemental Table 1.

*In vitro transcription of sgRNA*: Transcription reactions were conducted with HiScribe T7 High Yield RNA Synthesis Kit (NEB) according to the manufacturer's recommendations. Reactions proceeded at 37 °C overnight, and the sgRNA was subsequently purified with RNA Clean & Concentrator-5 Kit (Zymo Research). The concentration of sgRNA was determined by NanoDrop™ One/OneC (Thermo Fischer).

RNP preparation and electroporation were carried out following a protocol described by Ling, X. et al. [50] with minor modifications. Briefly, the RNP complex was prepared by gently mixing SpCas9 protein and sgRNA at a 1:3 molar ratio and then incubated at room temperature for 10 min. For electroporation, cells were harvested, washed with PBS buffer, and resuspended in electroporation buffer (Celetrix LLC) to a density of $2 \times 10^6$ cells/20 μl reaction system. The RNP complex was mixed with the cells and transferred to a 20 μl electroporation tube (Celetrix LLC). For RAW264.7 cells, 60 pmol RNP were electroporated under the optimized condition (520 V) using the electroporation machine CTX-1500A LE + (20–200 μl). Following electroporation, the cells were immediately transferred to a prewarmed medium for cultivation. SR-A knock-out single cells were

selected by fluorescence-activated cell sorting (FACS, Beckman MoFlo XDP), grown into clones, and evaluated for SR-A expression by western blot and flow cytometry analysis.

## Flow cytometry for IL-17A uptake

17Abp and SRAL mixture were prepared by mixing the stock solutions (800 µM) of 17 Abp and SRAL separately ($v/v = 1:1$) and preincubating overnight at 4 °C. Meanwhile, RAW264.7 was seeded into 48-well plates at 1 million cells per well. After that, cells were incubated with indicated treatments for the indicated time. After washing with PBS for three times, the cells were fixed and permeabilized by 4% polyformaldehyde (Solarbio) and the permeabilization buffer (eBioscience) respectively. Then, the cells were treated with the PE anti-mouse IL-17 antibody (1:200, Biolengend) for the flow cytometry analysis. The cells without treatment of peptides and IL-17A were used as the blank group during the flow cytometry analysis, while the cells treated with IL-17A only served as the control. The intracellular intensity was measured by the Beckman CytoFLEX LX flow cytometer.

To verify the binding of the IL-17A by mixed fi-17Abp and fi-SRAL, RAW264.7 cells were treated with 0.5 µg/ml IL-17A, 50 µM mixed fi-SRAL and fi-17Abp and IL-17RA at 0, 0.0125, 0.025, 0.05, 0.1, or 0.2 µg/ml for 6 h. Then, flow cytometry analysis was used to measure the intracellular IL-17A level according to the procedure mentioned above.

To verify the involvement of SR-A in the IL-17A endocytosis, RAW264.7 cells were treated with the SR-A inhibitor Poly I or the control Poly C at the indicated concentration together with the 50 µM 17Abp and SRAL mixture and 0.5 µg/ml IL-17A for 6 h. Then, flow cytometry analysis was used to measure the intracellular IL-17A level according to the procedure mentioned above.

To further verify the involvement of SR-A in the IL-17A endocytosis, wildtype RAW264.7 cells (WT) or SR-A knockout RAW264.7 cells (SR-A-KO) were incubated with 0.5 µg/ml IL-17A and 50 µM mixed fi-SRAL and fi-17Abp for 6 h. The uptake of the IL-17A by cells was analyzed through flow cytometry according to the procedure mentioned above.

To evaluate the uptake mechanism of IL-17A, RAW264.7 cells were pretreated with various inhibitors for 30 min at 37 °C: chlorpromazine (clathrin-mediated endocytosis inhibitor, 3 µg/ml), Me-β-cyclodextrin (0.5 mM, lipid rafting pathway inhibitors), EIPA (20 µM, pinocytosis inhibitor). In addition, the cells were cultured at 4 °C or treated with NaN₃ (100 mM) to assess the effect of energy on IL-17A uptake. After that, the cells were cultured with fresh medium containing inhibitors at the same concentrations, 50 µM 17Abp and SRAL mixture, and 0.5 µg/ml IL-17A for 6 h. The uptake of the IL-17A by cells was analyzed through flow cytometry according to the procedure mentioned above.

To confirm the involvement of lysosomes, RAW264.7 cells were pretreated with PBS, LPT (0.1 mg/ml), or BafA1 (100 nM) for 2 h at 37 °C. Subsequently, the cells were incubated with 50 µM 17A-Lytaca and 0.5 µg/ml IL-17A for 6 h with or without the presence of LPT or BafA1. The accumulation of IL-17A in the cells was then measured via flow cytometry, following the aforementioned procedure.

To investigate the impact of stoichiometry on IL-17A intake efficiency, RAW264.7 cells were treated with 17A-Lytaca at molar ratios of fm-17Abp to fm-SRAL at 1:4, 1:2, 1:1, 2:1, and 4:1, along with IL-17A protein for 6 h. The uptake of the IL-17A by cells was analyzed through flow cytometry according to the procedure mentioned above.

To study the influence of covalent peptide chimera linker length on IL-17A uptake, RAW264.7 cells were treated with 50 µM covalent peptides with different linker lengths, along with IL-17A for 6 h. The uptake of the IL-17A by cells was analyzed through flow cytometry according to the procedure mentioned above.

To evaluate the temporal IL-17A degradation, RAW264.7 cells were treated with 50 µM 17A-Lytaca and 0.5 µg/ml IL-17A for 6 h. After that, the cells were cultured with fresh medium for another 0, 3, 6, 12, 22 h.

The remaining IL-17A levels in cells were analyzed through flow cytometry according to the procedure mentioned above.

## Colocalization assay

RAW264.7 cells were plated on the glass coverslips (Solarbio) in 24-well plates for 1 day before the experiment. Cells were incubated with the peptide mixture and red fluorescent molecule RED-tris-NTA (Nanotemper) labled-IL-17A for 6 h. Cells were then fixed with 4% paraformaldehyde for 30 min, washed three times, permeabilized, and blocked with 0.3% Triton X-100 (Sangon Biotech) and 3% fat-free milk (Sangon Biotech) for 1 h. After the incubation with the primary antibody at 4 °C overnight, cells were then incubated with the appropriate secondary antibody and Hochest 33342 (Beyotime) at 37 °C for 30 min. Fluorescence images were captured by Nikon AXR confocal microscope using a Plan APO ×60, 1.42-NA oil objective with 405-nm violet laser, 488-nm blue laser, 561-nm green laser, and 640-nm red laser. Image analysis was performed with NIS-Elements Viewer and ImageJ software.

## Fluorescence resonance energy transfer (FRET)

N-terminal FITC-labeled 17Abp (fi-17Abp) and rhodamine B-labeled SRAL (rb-SRAL) mixture, fi-17Abp and acetyl-labeled SRAL (ac-SRAL) mixture, and acetyl-labeled 17Abp (ac-17Abp) and rb-SRAL mixture were prepared by incubating at 4 °C overnight at final concentrations of 400 and 400 µM, respectively. After that, the resulting peptide solution was examined at an excitation wavelength of 450 nm by a microplate reader (BioTek Synergy Neo2).

## Transmission electron microscopy (TEM)

To visualize the prepared peptides, 3 µl solution was loaded onto 200-mesh copper grids coated with carbon support films for 1 min. After removing the excess solution by blotting with filter paper, 3 µl of negative stain (2% aqueous phosphotungstic acid) was added and allowed to stain for 1 min. Following the removal of the remaining liquid with filter paper, the copper grids could dry on a hot fan and be directly imaged using a transmission electron microscope (JEM-1400 PLUS).

## Dynamic light scattering (DLS)

fm-17Abp and fm-SRAL or nf-TPP and nf-SRAL were mixed together at 4 °C overnight to form nanoparticles. Then the samples were centrifugated at 100×g for 10 min and the supernatant was pipetted out. The size distribution of the supernatant was analyzed by employing a Zetasizer instrument (Nano ZSP, Malvern Instruments).

## Time-dependent stability

fm-17Abp and fm-SRAL peptides were mixed together at 4 °C overnight to form nanoparticles and then stored in water at 4 or 37 °C. The hydrodynamic size and polymer dispersity index (PDI) were determined at 0, 2, 4, 6, 8, 10, and 12 h by DLS following the aforementioned procedure.

## 17A-Lytaca critical aggregation concentration measurement assay

The critical aggregation concentration of 17A-Lytaca was determined using dynamic light scattering (DLS) (Nano ZSP, Malvern Instruments). Solutions of 17A-Lytaca, ranging in concentration from 400 to 0.190 µM, were preincubated overnight at 4 °C. Each solution analysis was replicated three times. Scatter plots of intensity values of scattered light were subjected to linear regression to calculate the critical aggregation concentration (CAC).

## Surfactant competitive assay

17A-Lytaca was dissolved in H₂O or an 800 µM SDBS buffer at a concentration of 400 µM. The samples were then preincubated overnight,

followed by centrifugation at $100\times g$ for 10 min. The supernatant was pipetted out and analyzed by TEM, following the previously described procedure.

## Stoichiometry determination of 17A-Lytaca
The 17A-Lytaca co-polymer in varying molar ratios was separated from the peptide mixture through centrifugation ($100\times g$, 10 min). Subsequently, the supernatant was subjected to ultrafiltration for 10 min at $15,000\times g$ with an ultrafiltration device (Millipore10k, Merck) to remove the dissociative monomers. The resulting materials were dissolved in an acetonitrile/$H_2O$ solution and analyzed by HPLC, employing a standard curve based on peptide concentration.

## In vivo study
**Psoriasis model construction.** 8 to10-week-old female BALB/c mice were purchased from Peking University Health Science Center Department of Laboratory Animal Science. They were randomly divided into seven groups. Vehicle group (PBS administration) was treated with 62.5 mg/day of Vaseline cream while the other six groups were treated with 62.5 mg/day of 5% IMQ cream (Sichuan Med-shine Pharmaceuticals) on the shaved back skin (2 cm × 3 cm) of mice (days 1–7). They were saline groups (saline administration), MTX group (Sigma, 1.0 mg/kg/twice day), 17A-Lytaca group (17Abp and SRAL mixture were given 15 and 22 mg/kg/d after preincubated for 12 h), control-Lytaca group (fm-nbp and fm-SRAL mixture were given 12 and 22 mg/kg/d after preincubated for 12 h), fm-SRAL group (22 mg/kg/day), fm-17Abp group (15 mg/kg/day), all the mice were administrated with intraperitoneal injection.

**Measurement of skin inflammation severity.** The psoriasis area and severity index (PASI) consists of measurements of skin erythema, scale, and thickness. They were scored independently on a scale of 0–4: 0, none; 1, slight; 2, moderate; 3, marked; 4, very marked. The cumulative PASI score was obtained by adding the 3 index scores (scores of 0–12). The thickness of the mouse skin was measured using a micrometer, and the average value was measured three times a day. For histopathology analysis, tissues from the euthanized animals were fixed in formalin and embedded in paraffin. Sections (5 μm thickness) were stained with H&E.

**The gene expression level detection of cytokines.** *RNA extraction:* Collect the dorsal skin (100 mg) of the mouse. Carefully cut the tissue into 0.1 cm³ pieces, then homogenize with 1 ml of TRIzol (Thermo Fisher) using a grinder (Tiss-24, Jing Xin) to obtain the tissue suspension. Transfer 500 μl supernatant to a new EP tube, add 100 μl chloroform, and mix thoroughly. Then, the mixture was centrifuged at 4 °C at $12,000\times g$ for 15 min. Pipette the upper aqueous layer (around 300 μl) into a fresh tube and supplement with 300 μl of isopropanol, followed by $12,000\times g$ centrifugation at 4 °C for 10 min. Discard the supernatant and a white, solid mass of mRNA was observed on the bottom.

**cDNA synthesis.** Add 20 μl of RNAse-free water and incubate the tube in a water bath for 10 min to completely dissolve the mRNA. Quantify the mRNA concentration with NanoDrop (Thermo Fisher). Generate cDNA using the Hifair® III SuperMix Plus Kit (Yeason).

**Quantitative real-time PCR (qPCR) assay.** The cDNA was diluted 1:10 in RNase/DNAse-free water for quantitative real-time PCR (qRT-PCR). qRT–PCR reactions were run on a quantitative fluorescence PCR instrument (ABI QuantStudio 6) in triplicate. Hieff® qPCR SYBR Green Master Mix (Yeason) was used according to the manufacturer's instructions, and GAPDH was used as the housekeeping gene. The list of primer sequences is provided in Supplementary Table 1.

**The protein level detection of cytokines.** Plasma cytokine concentrations of IL-17A, IL-6, IL-22, and CCL-20 were determined by a commercial ELISA kit (Dogesce). For skin IL-17A detection, the back skin of mice was isolated and triturated with a grinder (Tiss-24, Jing Xin). Cell debris was precipitated via centrifugation ($1000\times g$, 4 °C, 15 min), and IL-17A was measured in the supernatant using the IL-17A ELISA kit (Dogesce) and normalized to the total protein concentration, which was determined via BCA assay.

## In vivo localization and clearance assay of 17A-Lytaca
To investigate the in vivo distribution and clearance of our co-polymer in psoriasis mice, 17A-Lytaca, comprising Cy5-labeled fm-17Abp and Cy7-labeled fm-SRAL, was administered via intraperitoneal injection to mice with induced psoriasis (treat with IMQ cream for six continuous days). Fluorescence imaging in both the Cy5 and Cy7 channels was conducted using the in vivo imaging system (IVIS Spectrum) at 4, 8, 12, 24, and 48 h post-injection. Further, for biodistribution assessment, mice were sacrificed at 6 h post-injection, and skins along with major organs (heart, liver, spleen, lung, and kidney) were removed for ex vivo fluorescence imaging.

## Serum clearance kinetics of 17A-Lytaca relevant peptides
To study the serum clearance kinetics properties of our co-polymer, 17A-Lytaca formed by Cy3-labeled fm-SRAL and Cy5-labeled fm-17Abp was administered to healthy mice through intraperitoneal injection. Serum samples were collected at 0, 1, 2, 3, 4, 8, 12, and 24 h for analysis. The fluorescence intensity of serum was measured with the in vivo imaging system (IVIS) spectrum.

## Western blot
Proteins were extracted using RIPA buffer (Beyotime) supplemented with protease inhibitors (Beyotime) and quantified using the BCA assay (Solarbio). SDS–PAGE was performed to separate proteins, which were then transferred to PVDF membranes (EMD Millipore). Next, the membranes were blocked in 5% fat-free milk for 2 h and incubated with primary antibodies at 4 °C overnight. including Rabbit anti-GAPDH antibody (1:5000, ab181602, Abcam), Mouse anti-GAPDH antibody (1:5000, 60004-1-Ig, Proteintech), Rabbit anti-CD2O4 antibody (1:1000, ab151707, Abcam), Mouse anti-PD-L1antibody (1:1000, 66248-1-Ig, Proteintech), Mouse anti-His-Tag antibody (1:2000, CW0286, CoWin Biosciences), Phospho-p44/42 MAPK (Erk1/2) (Thr202/Tyr204) (D13.14.4E) XP® Rabbit mAb, (1:1000, 4370T, Cell Signaling Technology), p44/42 MAPK (Erk1/2) (137F5) Rabbit mAb (1:1000 4695T, Cell Signaling Technology). After washing the membranes three times, secondary antibodies including HRP-conjugated Affinipure Goat Anti-Mouse IgG(H + L) (1:5000, SA00001-1, Proteintech), HRP-conjugated Affinipure Goat Anti-Rabbit IgG(H + L) (1:5000, SA00001-2, Proteintech) labeled with horseradish peroxidase (Proteintech) were applied at room temperature for 2 h and the immunoreactivity was visualized by supper ECL western blotting substrate (Coolaber). The densitometric analysis was performed by ImageJ software (NIH).

## Cell viability assay
Cell viability was measured using the CellTiter-Glo™ Luminescence Cell Viability Assay (Promega #G7570). This test allows the measurement of the amount of intracellular ATP, this metabolite being directly linked to the number of metabolically active cells. For this, 17A-Lytaca was prepared by mixing the stock solutions (800 μM) of fm-17Abp and fm-SRAL separately (v/v = 1:1) and preincubating overnight at 4 °C. Meanwhile, RAW264.7 was seeded into 96-well plates at 100,000 cells per well and grown for 12 h. The medium was replaced with different concentrations of 17A-Lytaca as 25, 50, 75, 100, and 125 μM in DMEM and the plates were incubated for a further 6 h period. After washing with DPBS, the medium was replaced by 100 μl DMEM per well. Then, 100 μl reagent was added into each well of 96-well plates using a multi-

channel pipette. Plates were incubated at room temperature on an orbital shaker for 10 min to stabilize the reaction, and then the luminescence signal of each well was recorded. All cell survival assays were performed in triplicates.

## Statistics

Data were analyzed by GraphPad Prism 9.4 software and given as mean ± SD. One-way analysis of variance (ANOVA) with Tukey's multiple comparisons test was used for statistical comparison among multiple (more than two) groups. Two-way ANOVA tests followed by Sidak's multiple comparisons test were conducted for a series of data collected at different time points and experiments that have two independent variables. The normality of data sets was assumed for ANOVA and was tested by Shapiro–Wilk tests. All statistical tests were unpaired. $P$ value of <0.05 was considered significant. Linear regression was used to determine the concordance.

## Reporting summary

Further information on research design is available in the Nature Portfolio Reporting Summary linked to this article.

## Data availability

All data generated or analyzed in this study are included in this article and its supplementary information files. Source data are provided with this paper.

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

## Acknowledgements

The authors are grateful for financial support from the National Natural Science Foundation of China (22177004, 92153301), and the State Key Laboratory of Natural and Biomimetic Drugs. We thank Professor Yiguang Wang (Peking University) for helpful discussions, and Drs. Jing Wang, Qian Wang, Guifang Duan, Xia Yuan, Lei Zhang, Xiaohui Zhang, and Yuan Wang (Peking University) for experimental assistance. BioRender.com was used in generating Supplementary Fig. 6f.

## Author contributions

S.D. supervised the entire project, obtained financial support, and approved the final version of the manuscript; Q.W. and S.D. conceived the study; Q.W., X.Y., and S.D. designed the experiments, analyzed the data, and wrote the manuscript. Q.W., X.Y., and R.Y. performed most of the experiments unless otherwise specified with the help of P.W., A.S., J.Z., H.L., and C.T.; Z.J. assisted with the flow cytometry analyses; W.L. assisted on the confocal microscopy.

## Competing interests
The authors declare no competing interests.
