## [Peer Review File · Nature Communications]

REVIEWER COMMENTS

Reviewer #1 (Remarks to the Author):

This manuscript presents a novel supramolecular co-assembly system called LYTACA, which utilizes self-assembling peptides with FITC or Fmoc tags to bind scavenger receptor A (SR-A) and targeted proteins simultaneously, leading to targeted protein degradation. Two LYTACAs were successfully developed to induce lysosomal degradation of extracellular protein IL-17A and membrane protein PD-L1 in RAW264.7 cells, with supportive data indicating that the formation of a supramolecular complex plays a crucial role in the targeted degradation of IL-17A. In vivo data suggests that this supramolecular co-assembly system exhibits higher efficacy compared to IL-17A antibodies. The authors further validate the versatility of this strategy by developing another LYTACA that degrades PD-L1. Overall, this study demonstrates the potential of LYTACAs in inducing the lysosomal degradation of proteins, presenting a promising approach for targeted protein degradation. However, there are additional questions that need to be addressed in order to publish this work in Nature Communications.

1. The degradation platform provides a strategy to degrade extracellular and cell surface protein, but due to its dependency on SR-A, cell-surface targets are limited only on certain immune cells like macrophage. To evaluate the universality of this technology, it's necessary to test if the system also works in other immune cells, such as T cells and B cells.
2. For Fig 1D, if the IL17A is overdosed in the medium, the time-course result should not be interpreted as IL17A undergoes degradation in cell, as there is always plenty of IL17A in the medium entering the cells. Could it be a sign of SR-A depletion? Evidence of IL17A level in the medium, as well as SR-A abundance change over time, should be provided.
3. For Fig 2, an additional competition experiment from IL17A side is encouraged.
4. The FRET assay is inconclusive because the decrease of FRET at 530nm could be attributed to random collisions rather than the formation of co-assembled supramolecular complexes. Therefore, it is necessary to include a blank control, which is without FITC labeling.
5. Although authors conducted FRET and transmission electron microscopy to support the formation of supramolecule. It's encouraged to do a surfactant competitive assay since surfactants are known to be capable of interacting with the aromatic rings, disrupting the self-assembly process.
6. Why both peptides do not form homo-polymer of the same size as the mixed peptide? Is there any method to determine the stoichiometry of the co-polymer? How would the stoichiometry affect the intake efficiency? Authors should investigate effect of the ratio of the two peptides.
7. For Fig 3e, how long does it take to completely degrade the IL17a in the lysosome? And to further validate the degradation is dependent on lysosome, one more lysosome inhibitor targeting different processes of lysosome pathway, such as BafA1, should be added.
8. Fig 4g caption, day 3 or day 4? The number is not consistent between the figure and the caption.

9. For Fig 3f and supplementary fig s, the SR-A level in DC2.4 is lower than U-118MG and HaCaT, however, IL-17A uptake of DC2.4 is higher than other two cell lines. The authors should explain this contradiction.

10. Supplementary fig 3 and fig 4 are cited wrong in the article.

11. The failure of covalent chimera, fi-SRAL-17Abp may be caused by the short linker between two peptides. Do authors screen the linker length?

Reviewer #2 (Remarks to the Author):

In this manuscript, Wang et al. discover that a supramolecular assembly of peptides that bind IL-17A and SR-A lead to uptake of IL-17A into macrophages, eventually resulting in degradation of IL-17A. The authors attempt a similar approach to degrade membrane-bound PD-L1. The authors attempt in vivo demonstration of IL-17A removal in a mouse model of psoriasis. This approach is conceptually the same as LYTACs and KINETACs, in that they suggest co-engagement of an internalizing receptor with a target protein. The advance in this work is that, potentially, it is possible to mix binders of a cell surface receptor and target non-covalently leading to target generation. Overall, the work requires more controls, more discussion of the nature of the complex they use for degradation, and more demonstration of clear degradation in vivo. Additionally, the paper should be amended to be precise about the actual advances of the work and their place in the literature, rather than drawing vague conclusions that issues which plague PROTACs also impact extracellular degraders.

Comments on Abstract and Introduction:

The authors mention linkers and attachment of degraders as a key area of need, however, they do not provide evidence that the same linker effects that impact PROTACs also impact extracellular degraders like LYTACs or PROTABs. If there is no evidence to support these assumptions, the authors should amend their abstract and introduction to not mention a problem that is not a limitation in the field.

The authors mention they develop a novel TPD approach, however, this is an imprecise statement. The authors have developed a method for generating an extracellular degrader that operates by the same mechanism as other degraders, but is a slightly different molecular composition. This is the equivalent of generating a new linker structure for a PROTAC and calling it a new approach. If the authors believe their mechanism is novel (not just the receptor or ligand), further evidence to support this claim is needed.

Comments on Main Text:

The authors use a combination of inhibitors and treatment conditions to draw the conclusion that SR-A is the receptor which is targeted. However, they also not a contribution of pinocytosis. These experiments are not conclusive in terms of identifying the mechanism of uptake, thus, knockout experiments for SR-A are needed (either CRISPR or siRNA with blot evidence and sequencing). Without these experiments, no mechanistic conclusion can be reached and the work is substantially reduced. The references to multiple cell lines with different levels of SR-A is not evidence of SR-A involvement, merely differential uptake ability of different cell lines.

FITC imaging is used in Fig. 2d, however, FITC is well-known to be quenched in acidic environments, and LysoTracker is not a specific marker for lysosomes. LAMP1 imaging is more accurate for claiming colocalization with lysosomes and should be performed here.

The authors have left out key control experiments and comparisons for their in vivo data. A major component which needs to be addressed is treatment with the SRAL peptide alone. Without this control, it is not clear if a general macrophage activation strategy with increased inflammation underlies their observed effect with the combination of SRAL and 17Abp. The authors should directly compare the 17A-LYTACA to 17-Abp for clearance of IL17-A in Fig. 4g and 4h. The effect size difference is small, if significant at all, and thus is unclear if clearance of IL17-A with a Lytaca is actually occurring in vivo or underlies the observed pathology. Indeed, without the SRAL peptide treatment alone, the cytokine mRNA profiling is preliminary at best. P values were calculated using a t-test, but the more statistically relevant method would be with an ANOVA across all groups. Overall the in vivo data is not convincing, especially without controls or adequate statistically comparisons. Further, adequate PK experiments must also be presented, such as clearance time of the relevant peptides from serum, and if the conjugate is capable of reaching the relevant location given the skin depletion data. Additionally, what evidence exists that the peptide complexes going to remain in vivo?

For the PD-L1 degradation experiments in Fig. 5, the authors must also include mechanistic experiments which demonstrate their peptides form similar complexes, and are utilizing the SR-A receptor through knock-out experiments. Further, quantification of the Western blot degradation results in Fig. 5d in biological triplicate in bar graph form is the standard of the field. Additionally, evidence that SR-A is recycling, as their diagram in Fig. 5a demonstrates should be presented in the form of Western blots of degradation experiments.

Comments on Discussion: Overall, not enough evidence is presented to suggest this is a general approach for peptide assembly. While the self-assembled structures are characterized in vitro, it is unclear exactly what drives their assembly. For this to be a new class of extracellular degraders, there must be some logic for assembly of any target binder into these complexes. However, the lack of characterization of the PD-L1 Lytaca, and the unclear explanation for self-assembly of the IL-17 molecule makes it difficult to judge if these complexes are the same for different targets. The comment "In addition, when compared to covalent chimeras that demand a strenuous optimization process, this co-

assembly degradation system is relatively straightforward to prepare using diverse peptide binders" does not make sense. There is no evidence that strenuous linker optimization efforts must be applied to extracellular binders, and the rate limiting step for any degrader is the necessity of generating a peptide binder in the first place. Overall, significantly more work must be done to rigorously demonstrate the nature of the complexes, their generality, and their specificity.

Reviewer #3 (Remarks to the Author):

The manuscript by Wang et al. introduces a methodology for the targeted degradation of extracellular proteins, building upon the principles of the LYTAC approach used to target surface receptors. However, this new approach, called lysosome-targeting co-assemblies (LYTACA), has notable differences. Firstly, it utilizes class A scavenger receptors (SR-A) to direct cargo toward lysosomal degradation. Secondly, it employs two separate peptides that can assemble together.

The study provides evidence that Lytaca targeting IL-17A promotes the internalization of IL-17A by the RAW264.7 macrophage cell line. Furthermore, the authors show that IL-17A-Lytaca can partially reduce disease progression in an imiquimod-induced psoriasis mouse model. Additionally, the authors demonstrate the ability of Lytaca particles that binds PD-L1 to target and downregulate surface PD-L1 on RAW264.7 macrophages.

However, a key conceptual issue with this proposed approach arises from the rapid internalization of the SR-A-targeting peptide (as depicted in Figure 1). Based on this observation, it is likely that LYTACA particles would be swiftly cleared from circulation, even if they fail to bind the presumed cargo proteins like IL-17A. This would limit the system's ability to remove IL-17A that is continuously released at the site of inflammation. The only conceivable solution would be to administer large amounts of IL-17A-Lytaca on a regular basis – presumably daily, as shown in the in vivo experiment (Figure 4). This seems very disadvantageous for use in clinical settings.

In contrast, targeting surface proteins like PD-L1 may offer certain advantages since the internalized LYTACA particles would be more likely to associate with the target protein. Nevertheless, even under most optimal conditions using the RAW264.7 cell line, adding PD-L1-Lytaca leads to only a 30% decrease in surface PD-L1, as shown in Figure 5. This level of degradation is unlikely to be relevant in cancer settings, particularly considering that PD-L1 would presumably only be degraded on SR-A-expressing cells. Consequently, the presented LYTACA method does not appear to be very promising at the current stage.

Several key points to be addressed:

1. The authors demonstrate that IL-17A-Lytaca particles can promote the internalization of IL-17A. However, it is important to determine the extent to which this internalization depletes the provided IL-17A from the medium to evaluate the proposed approach's efficiency.

2. It is possible that upon binding to SR-A, the Lytaca particles can directly activate a signaling response. Therefore, it is necessary to investigate whether the treatment of cells with LYTACA leads to cellular activation.

3. Related to the previous point, the psoriasis experiment in Figure 4 is missing control to rule out the possibility that IL-17A-Lytaca particles trigger SR-A-mediated activation of signaling responses, which might be independent of IL-17 depletion. The author should prepare Lytaca particles with a control peptide that does not bind to IL-17A and confirm that the administration of this control-Lytaca does not result in the observed effects in the in vivo experiment.

Minor points:

1. In several figures, such as Figure 3e, or 5b-c, the y-axis is segmented into two parts without an apparent reason. It seems that the purpose is solely to visually emphasize the differences. It would be more appropriate to present these graphs with a non-segmented y-axis.

2. The effect of Lytaca on psoriasis progression, as shown in Figures 4b and 4c, should be accompanied by statistical analysis. The observed effect seems to be weak, and conducting statistical tests will provide a more robust evaluation.

3. The results presented in Figure 4e, demonstrating an increase in skin thickness, should be quantified and displayed in a separate graph.

4. The description of the statistical analysis is lacking. It is unclear why the authors used one-way ANOVA to compare two samples in Figures 1, 2, 3, and 5, as this test is typically employed for comparing more than two groups. In contrast, a t-test is used in Figure 4. The methods section should provide an explanation for the choice of specific statistical tests. Additionally, information about how the normality of the data was determined, especially if using tests that assume a normal distribution, should be included.

5. Some of the statistical analyses yield surprising results. For example, in Figure 3f, there is a very clear distinction between the internalization of IL-17 by HaCaT cells in the presence or absence of IL-17A-Lytaca. However, the authors state that it is not statistically significant. This raises questions about the appropriateness of the statistical tests used.

Reviewer #4 (Remarks to the Author):

In this paper, Wang et al developed self-assembled structure of two binding proteins. They co-assembled SR-A peptide ligand and IL-17A binding domain. The resulting assembled structure efficiently removed extracellular IL-17A by lysosomal degradation in both cell lines and mouse models. In addition, the authors applied this system to remove PD-L1 receptor on cell surface, not in extracellular environment. I think that this study is meaningful, but premature for publication in my opinion because key data is missing. I recommend the publication after major revision as below.

1. What is the main reason of the increased effect of self-assembled structure? It is the key point and should be determined by experimental data and sufficient discussion. In addition, why did the covalent fi-SRAL-17Abp chimera show no increase of IL-17A uptake in cells?
2. In studies of self-assembled structure, its time-dependent stability and critical concentration for self-assembly are important information and should be provided.
3. In Fig3g, the structure of 17A-Lytaca is too small and hard to figure out the structure.
4. In mouse experiments, I cannot find the critical advantage of 17A-Lytaca over MTX. Is the difference in Fig 4c between the two groups significant? Fig 4h and CCL-20 showed better effect of MTX.
5. In 'Notably, the 17A-Lytaca treatment did not show any obvious toxicity to the organs of mice (Supplementary Fig. 4)', the organ toxicity data is Fig S5, not S4.
6. It would be better to perform PD-L1 degradation study in cancer cells, not RAW264.7.
7. In development of new molecules, concentration-dependent cell viability data is essential.

Author Responses to Reviewer Comments

Reviewer #1.

1. The degradation platform provides a strategy to degrade extracellular and cell surface protein, but due to its dependency on SR-A, cell-surface targets are limited only on certain immune cells like macrophage. To evaluate the universality of this technology, it's necessary to test if the system also works in other immune cells, such as T cells and B cells.

Response. We appreciate the reviewer's suggestions. To assess the universality of our strategy, we extended its application to cancer cells, where SR-A is highly expressed. Our results demonstrate that LYTACAs effectively degrade PD-L1 in both A549 and HepG2 cell lines (new Fig. 5d, new Supplementary Fig 11c, 13b, and 13c). Accordingly, the text, "Given that scavenger receptors (SRs) are highly expressed on cancer cells^{13, 40, 41}, we chose A549 and HepG2 cells to evaluate the capability of our LYTACA in degrading PD-L1 in tumor cells. Western blot results revealed successful degradation in both cell lines, although the efficiency was relatively low, with approximately 50% degradation achieved in HepG2 cells (Fig. 5d, Supplementary Fig. 11c). ... Remarkably, compared with fm-PD-L1-Lytaca, nf-PD-L1-Lytaca exhibited significantly enhanced degradation efficiency in both macrophages (Fig. 5g) and cancer cells (Supplementary Fig. 13b, c)", has been added to describe the results, and these changes have been incorporated into the manuscript on Pages 11-12.

Moreover, to showcase the versatility of our LYTACA strategy across different receptors, we investigated the asialoglycoprotein receptor (ASGPR). ASGPR is primarily expressed on hepatocytes and requires multivalent ligand binding (e.g., tri-GalNAc) for activation. Using a newly designed PD-L1-LYTACA equipped with ASGPR-targeting mono-GalNAc and PD-L1-targeting TPP warheads, we successfully reduced PD-L1 levels in on HepG2 cells (new Fig. 5j, k). This result not only demonstrates that our LYTACA strategy extends beyond immune cell receptors but also underscores a distinct advantage: a readily accessible mono-valent structure can be manipulated to effectively interact with receptors that typically demand complex multivalent structures, thus eliminating the need for laborious synthesis and optimization (*Nat. Chem. Biol.* **2021**, *17*, 947-953; *ACS Cent. Sci.* **2021**, *7*, 499–506). To explain this further and present the experimental results, we added text to the manuscript (Page 12) stating, "To showcase the versatility of our LYTACA strategy across different receptors, we investigated the asialoglycoprotein receptor (ASGPR), primarily expressed on hepatocytes, which recognizes glycoproteins bearing N-acetyl galactosamine (GalNAc) or galactose (Gal)^{45, 46, 47}. Previous studies indicate that interactions with a degrader containing multivalent ligands, such as tri-GalNAc, can enable ASGPR to facilitate lysosome-targeting protein degradation^{6, 10, 11}. In this study, we employed a newly designed PD-L1-Lytaca equipped with ASGPR-targeting mono-

GalNAc and PD-L1-targeting TPP warheads (Fig 5j). The successful reduction of PD-L1 levels in HepG2 cells (Fig 5k) not only demonstrates that the LYTACA strategy extends beyond the SR-A receptor but also underscores a distinct advantage: the effect of a readily accessible mono-valent structure can be amplified to achieve effective interaction with receptors that typically demand complex multivalent ligands.”

Figure. 5d Dose escalation experiment for PD-L1 degradation in A549 cells treated with 10, 25, 50 and 100 μM fm-PD-L1-Lytaca or 50 μM fm-TPP/fm-SRAL for 12 h.

Figure. S11c Western blot of PD-L1 in HepG2 cells treated with 50 μM fm-TPP, fm-SRAL or fm-PD-L1-Lytaca for 12 h.

Figure. S13b Western blot of PD-L1 in HepG2 cells treated with 50 μM nf-TPP, nf-SRAL or nf-PD-L1-Lytaca for 12 h.

Figure. S13c Western blot of PD-L1 in A549 cells treated with 50 μM nf-TPP, nf-SRAL or nf-PD-L1-Lytaca for 12 h.

Figure. 5j Schematic illustration of ASGPR binding with Tri-GalNAc or fm-PD-L1-Lytaca consist of TPP ligand and mono-GalNAc ligand.

Figure. 5k Western blot of PD-L1 in HepG2 cells treated with 50 μ M fm-TPP, ASGPRL or PD-L1-Lytaca (fm-TPP+ASGPRL) for 12 h.

2. For Fig 1D, if the IL17A is overdosed in the medium, the time-course result should not be interpreted as IL17A undergoes degradation in cell, as there is always plenty of IL17A in the medium entering the cells. Could it be a sign of SR-A depletion? Evidence of IL17A level in the medium, as well as SR-A abundance change over time, should be provided.

Response. To investigate whether the decline in IL-17A signal during the time-course experiment results from SR-A depletion, we assessed the levels of both IL-17A in the medium and SR-A on the cell surface over time following the treatment of RAW 264.7 cells with 17A-Lytaca and IL-17A for the specified duration.

The western blot results indicated that the level of IL-17A in the medium did not exhibit observable changes (Fig. R1a, b), suggesting an overdose of the provided protein. Simultaneously, we monitored changes in SR-A abundance through flow cytometry. The results showed that SR-A recovered at 48 h following a decline at 24 h (Fig. R1c), which does not support the hypothesis of SR-A depletion.

Consequently, we propose that the decline in IL-17A signal, as illustrated in the new Fig. 1d, may be linked to the consumption and reduced concentration of our 17A-Lytaca during the latter stage of the time-course experiment. However, in clinical settings, the reported concentration of IL-17A in psoriatic lesions is 0.01 ng/ml (*Sci. Rep.* **2016**, *6*, 26071; *J. Immunol.* **2006**, *177*, 36–39), significantly lower than the concentration (0.5 μ g/ml) we used in our in vitro uptake assay. Therefore, the efficiency of LYTACA may be sufficient to achieve the degradation of IL-17A at levels relevant to those observed in patients.

Figure. R1a Medium IL-17A measurement by western blot in the time-course experiment for IL-17A uptake. RAW264.7 cells incubated with 0.5 $\mu\text{g/ml}$ IL-17A and 50 μM mixed fi-SRAL and fi-17Abp for 1, 3, 6, 12, 24 and 48 h. Medium IL-17A was taken at the indicated time and analyzed by western blot.

Figure. R1b The corresponding quantified result of **Fig. R1a**. Densitometry was used to calculate protein levels, and data ($n = 3$ biologically independent experiments) were normalized to the GAPDH.

Figure. R1c Cell surface SR-A receptor quantification via flow cytometry in the time-course experiment for IL-17A uptake. RAW264.7 cells incubated with 0.5 $\mu\text{g/ml}$ IL-17A and 50 μM mixed fi-SRAL and fi-17Abp for 1, 3, 6, 12, 24 and 48 h. SR-A was measured at the indicated time.

Figure. 1d Time-course experiment for IL-17A uptake in RAW264.7 cells incubated with 0.5 $\mu\text{g/ml}$ IL-17A and 50 μM mixed fi-SRAL and fi-17Abp for 1, 3, 6, 12, 24 and 48 h.

3. For Fig 2, an additional competition experiment from IL17A side is encouraged.

Response. In addition to the inhibition experiments using SR-A inhibitor Poly-I, we have conducted an additional competition experiment using IL-17RA, which competes for the same binding domain of IL-17A with 17Abp. The results showed that the LYTACA-induced endocytosis of IL-17A was inhibited by IL-17RA in a concentration-dependent manner (new Fig. 1e).

These results are detailed in the manuscript on Page 5: “To investigate the mechanism underlying the endocytosis of IL-17A induced by the peptide mixture of fi-SRAL and

fi-17Abp, we performed competitive inhibition assays from both IL-17A and SR-A binding perspectives. Since peptide 17Abp competes for the same binding domain of IL-17A with IL-17RA^{15, 23}, we employed IL-17RA as a competitor. Cells were treated with varying concentrations of IL-17RA in the presence of fi-SRAL and fi-17Abp mixture, along with IL-17A. The intracellular IL-17A signal exhibited a concentration-dependent attenuation contingent on IL-17RA levels (Fig 1e). Moreover, an additional competition assay was performed using a SR-A competitive inhibitor, polyinosinic acid (poly I), and a non-inhibitor control, polycytidylic acid (poly C)²⁰.”

Figure. 1e IL-17A binding competition experiment for IL-17A uptake in RAW264.7 cells treated with 0.5 µg/ml IL-17A, 50 µM mixed fi-SRAL and fi-17Abp, IL-17RA at 0, 0.0125, 0.025, 0.05, 0.1 or 0.2 µg/ml for 6 h.

4. The FRET assay is inconclusive because the decrease of FRET at 530nm could be attributed to random collisions rather than the formation of co-assembled supramolecular complexes. Therefore, it is necessary to include a blank control, which is without FITC labeling.

Response. We repeated the FRET assay with a more rigorous control: two peptides, ac-17Abp and ac-SRAL, lacking FITC and Rhodamine B labeling were prepared using an acetyl group to cap their N-terminal (new Fig. 2a). The experimentation results revealed that the mixture of fi-17Abp and ac-SRAL exclusively displayed the characteristic FITC emission spectrum (purple), while the mixture of ac-17Abp and rb-SRAL exhibited only the characteristic RB emission spectrum (green). In contrast, the mixture of both fluorescently labeled peptides, fi-17Abp and rb-SRAL, displayed two characteristic peaks indicative of FRET signals (red). Combining these results, we conclude that the decrease in the fluorescent signal at 530 nm is due to the proximity of FITC and RB groups in the formed co-assembling supramolecules, ruling out the possibility of weakened signals caused by random collisions.

Accordingly, the corresponding texts on Page 6-7 has been revised to: “Excited by a 450 nm wavelength fluorescence, the mixture of fi-17Abp and non-fluorescent ac-SRAL displayed a strong fluorescence peak at 530 nm (characteristic emission of FITC),

while the mixture of ac-17Abp and rb-SRAL showed a weak fluorescence peak at 595 nm (characteristic emission of RB). For the mixture of fi-17Abp and rb-SRAL, the fluorescence at 530 nm was weakened while that at 595 nm was increased sharply, indicating FRET occurrence and the potential formation of co-assembled supramolecular complex. The FRET signal of two peptides was also observed in RAW264.7 cells (Supplementary Fig. 4b), further corroborating the conclusion.”

Figure. 2a Fluorescent emission spectra of the mixed fi-17Abp and rb-SRAL (red), the mixed fi-17Abp and ac-SRAL (purple curve), and the mixed ac-17Abp and rb-SRAL (green). The peptides were mixed in a 1:1 molar ratio, and the excitation wavelength is 450 nm. ac-SRAL: N-acetylated SRAL. ac-17Abp: N-acetylated 17Abp.

Figure. S4b Visualization of FRET phenomenon of peptide fi-17Abp and rb-SRAL in RAW 264.7 cells by confocal microscopy. Scale bar: 5 μ m.

5. Although authors conducted FRET and transmission electron microscopy to support the formation of supramolecule. It's encouraged to do a surfactant competitive assay since surfactants are known to be capable of interacting with the aromatic rings, disrupting the self-assembly process.

Response. In response to the reviewer’s suggestion, we opted for sodium dodecylbenzenesulfonate (SDBS), containing a benzene ring, as a competitor in the surfactant competitive assay. The TEM image revealed that the addition of SDBS perturbed the original morphology and size of 17A-Lytaca (new Supplement Fig. 4c). We reason that the benzene moiety of SDBS may interact with the aromatic rings of the Fmoc on the peptide N-terminus, thereby disrupting the co-assembly process of the two peptides. To convey this result, the manuscript (Page 7-8) now includes the following text: “To gain more insight into the nature of the peptide co-assembly, we conducted a surfactant competitive assay²⁸ using SDBS (Sodium dodecylbenzene sulfonate) as a competitor. The TEM image showed that the mixture of fm-SRAL and fm-17Abp in SDBS-containing buffer could not form the same spherical nanoparticles as it did in water (Supplementary Fig. 4c). This disruption of co-assembly is likely

caused by interactions between the aromatic moiety of SDBS and the Fmoc groups, supporting the aromatic-driven self-assembly process.”

Figure. 4c Representative TEM images for Mixed fm-SRAL and fm-17Abp in H₂O or sodium dodecylbenzene sulfonate (SDBS, 800 μ M) buffer at a concentration of 400 μ M. Scale bar, 100 nm.

6. Why both peptides do not form homo-polymer of the same size as the mixed peptide?

Response. We hypothesize the distinctive charge distribution of the two peptides impedes formation of homopolymers of comparable size to that observed in the mixed peptide. For instance, SRAL, being polyanionic, may be constrained in size due to charge repulsion. The introduction of 17Abp, however, is anticipated to alter the charge distribution of SRAL, mitigating repulsive forces and consequently facilitating the formation of co-assembled supramolecular structures.

7. Is there any method to determine the stoichiometry of the co-polymer? How would the stoichiometry affect the intake efficiency? Authors should investigate effect of the ratio of the two peptides.

Response. To determine the stoichiometry of the co-polymer, we employed high-performance liquid chromatography (HPLC) analysis. Following a 12-hour incubation of the peptide mixture in varying molar ratios, samples were centrifuged (1000 rpm, 10 min) to remove precipitates. The supernatant was then subjected to ultrafiltration (10 min at 15000g using an ultrafiltration device - Millipore 10k, Merck) to eliminate unassembled monomers. The resulting materials were dissolved in an acetonitrile/H₂O solution and analyzed by HPLC, employing a standard curve based on peptide concentration (Fig. R2a). For 17A-Lytaca with molar ratios of fm-17Abp to fm-SRAL at 1:4, 1:2, 1:1, 2:1, and 4:1, the calculated assembly ratios were approximately 11:1, 6:1, 7:1, 26:1, and 66:1, respectively (Fig. R2b).

To investigate the impact of stoichiometry on IL-17A intake efficiency, RAW264.7 cells were treated with 17A-Lytaca at molar ratios of fm-17Abp to fm-SRAL at 1:4, 1:2, 1:1, 2:1, and 4:1, along with IL-17A protein for 6 hours. An excess of fm-SRAL over fm-17Abp had a similar effect on IL-17A uptake compared to equal molar addition of fm-SRAL and fm-17Abp. Conversely, when the added quantity of fm-SRAL was

less than that of fm-17Abp in preparing 17A-Lytaca, endocytosis efficiency significantly decreased (Fig. R2c). These results suggest that an adequate amount of fm-SRAL in 17A-Lytaca is essential for achieving effective IL-17A endocytosis.

To clarify this point, we added the following text to page 8 of the manuscript: “Furthermore, we investigated the composition of the co-assembled complex under conditions of mixing fm-SRAL and fm-17Abp in a molar ratio of 1:1. After a 12-hour incubation, the sample underwent centrifugation to remove the aggregates and subsequent ultrafiltration to eliminate the residual peptides. High-performance liquid chromatography (HPLC) analysis of the resulting materials revealed a stoichiometry of approximately 1:7 for fm-SRAL and fm-17Abp (Supplementary Fig. 4f).”

Figure. R2a Schematic illustration of the work flow about stoichiometry determination.

Figure. R2b HPLC result of the co-polymer with molar ratios of fm-17Abp to fm-SRAL at 1:4 (160 μ M: 640 μ M), 1:2 (267 μ M: 533 μ M), 1:1 (400 μ M: 400 μ M), 2:1 (533 μ M: 267 μ M), and 4:1 (640 μ M: 160 μ M).

Figure. R2c Ratio effect assay for IL-17A uptake in RAW264.7 cells incubated with 0.5 μ g/ml IL-17A and 17A-Lytaca with molar ratios of fm-17Abp to fm-SRAL at 1:4 (160 μ M: 640 μ M), 1:2 (267 μ M: 533 μ M), 1:1 (400 μ M: 400 μ M), 2:1 (533 μ M: 267 μ M), and 4:1 (640 μ M: 160 μ M).

9. For Fig 3e, how long does it take to completely degrade the IL17a in the lysosome?

Response. To figure out how long it takes to completely degrade the IL-17A, we conducted a time-dependent IL-17A degradation assay. After treating the cells with 17A-Lytaca and IL-17A for 6 h, the culture medium was refreshed without the addition of 17A-Lytaca and IL-17A. Subsequently, we assessed the intracellular IL-17A signal

in cells at 0 h, 3 h, 6 h, 12 h, and 22 h using flow cytometry. The results showed that approximately 22 hours were needed to achieve complete degradation of IL-17A (new Fig. 3d). This information has been incorporated into the manuscript on page 8 as follows: “Moreover, after a 6-hour treatment with IL-17A and 17A-Lytaca, followed by replenishing the culture medium without these components, a time-dependent reduction of the endocytosed IL-17A signal was observed, indicating a complete degradation of the endocytosed IL-17A in macrophage cells over approximately 22 h (Fig 3d).”

Figure. 3d Temporal IL-17A degradation experiment. The intracellular IL-17A signal was assessed by flow cytometry after a 6 hour-treatment with 0.5 $\mu\text{g/ml}$ IL-17A and 50 μM 17A-Lytaca.

10. And to further validate the degradation is dependent on lysosome, one more lysosome inhibitor targeting different processes of lysosome pathway, such as BafA1, should be added.

Response. Experiments using two lysosome inhibitors, LPT and BafA1, were conducted. The results indicate that both the LPT- and BafA1-treated groups showed significantly higher IL-17A accumulation than that without lysosome inhibitor treatment (new Fig. 3e). To describe this result, texts have been added to the manuscript (Page 8) as follows: “To further confirm the involvement of lysosomes in the 17A-Lytaca-promoted IL-17A degradation, cells were treated with the lysosome inhibitor Leupeptin (LPT) and Bafilomycin A1 (BafA1). Both inhibitor-treated groups exhibited significantly higher IL-17A accumulation in cells, indicating a suppression of lysosomal degradation (Fig 3e).”

Figure. 3e Lysosome inhibition assay. Cells were incubated with 0.5 μ g/ml IL-17A and 50 μ M 17A-Lytaca for 6 h in the presence or absence of 100 nM BafA1 or 0.1 mg/ml leupeptin (LPT). The accumulation of IL-17A was detected via flow cytometry.

11. Fig 4g caption, day 3 or day 4? The number is not consistent between the figure and the caption.

Response. Thank the reviewer for pointing out this error. We have re-conducted the animal experiments and this issue has been corrected in this revised manuscript.

12. For Fig 3f and supplementary fig s, the SR-A level in DC2.4 is lower than U-118MG and HaCaT, however, IL-17A uptake of DC2.4 is higher than other two cell lines. The authors should explain this contradiction.

Response. We hypothesized that DC2.4 cells, serving as antigen-presenting cells (APCs) responsible for capturing, processing, and presenting antigens to immune cells (*Cell Res.* **2017**, *27*, 74-95; *Cell.* **2001**, *106*, 255-258), might have higher uptake capability compared to U-118 MG and HaCaT cells. This may be related to other endocytosis pathways other than the SR-A-dependent mechanism. Therefore, the results from this assay are better read as supporting the cell selectivity of 17A-Lytaca, rather than correlating the uptake capability solely with the expression level of SR-A. We have made suitable revisions in the main text to clarify this point.

13. Supplementary fig 3 and fig 4 are cited wrong in the article.

Response. Corrections have been made.

14. The failure of covalent chimera, fi-SRAL-17Abp may be caused by the short linker between two peptides. Do authors screen the linker length?

Response. In addition to the covalent peptide chimera (peptide C6) conjugated by 6-aminocaproic acid, we have also evaluated two additional covalent chimeras (C9 and

C12) with varying the lengths of linkers (**Fig. R3a**). The result showed that neither of these covalent chimeras could promote the uptake of IL-17A (**Fig. R3b**). This may be explained by the intrinsic property of SR-A, which demands multivalent binding afforded by a branched polyanionic ligand, such as the reported polymer-based structure (*Mol. Pharm.* **2020**, *17*, 3794-3812). Therefore, the single peptide in the covalent chimera may not be sufficient to bind SR-A.

Figure. R3a Structures of designed covalent chimeras with various conjugate linkers.

Figure. R3b Experiment for linker length screening. RAW264.7 cells were exposed to 0.5 μg/ml IL-17A and 50 μM each of peptides C6, C9, and C12. Intracellular IL-17A signal was evaluated using flow cytometry.

Reviewer #2.

1. In this manuscript, Wang et al. discover that a supramolecular assembly of peptides that bind IL-17A and SR-A lead to uptake of IL-17A into macrophages, eventually resulting in degradation of IL-17A. The authors attempt a similar approach to degrade membrane-bound PD-L1. The authors attempt in vivo demonstration of IL-17A removal in a mouse model of psoriasis. This approach is conceptually the same as LYTACs and KINETACs, in that they suggest co-engagement of an internalizing receptor with a target protein. The advance in this work is that, potentially, it is possible to mix binders of a cell surface receptor and target non-covalently leading to target generation. Overall, the work requires more controls, more discussion of the nature of the complex they use for degradation, and more demonstration of clear degradation in vivo.

Comments on Abstract and Introduction:

The authors mention linkers and attachment of degraders as a key area of need, however, they do not provide evidence that the same linker effects that impact PROTACs also impact extracellular degraders like LYTACs or PROTABs. If there is no evidence to support these assumptions, the authors should amend their abstract and introduction to not mention a problem that is not a limitation in the field.

Response. We greatly appreciate the suggestions from Reviewer #2. Researchers had reported that the length of the linker in LYTAC degraders is crucial for target degradation, necessitating thorough screening (*ACS Chem. Biol.* **2023**, *19*, 1611-1623;). For example, Fang *et al.* demonstrated that variations in linker length could lead to a significant difference in PD-L1 degradation efficiency, spanning from 16% to 63% (*J. Am. Chem. Soc.* **2022**, *144*, 21831-21836). This observation is consistent with findings in PROTAC cases (*Nat. Rev. Clin. Oncol.* **2023**, *20*, 265-278; *Eur. J. Med. Chem.* **2020**, *201*, 112451). To provide further support for this point, we have included relevant references in the manuscript.

The authors mention they develop a novel TPD approach, however, this is an imprecise statement. The authors have developed a method for generating an extracellular degrader that operates by the same mechanism as other degraders, but is a slightly different molecular composition. This is the equivalent of generating a new linker structure for a PROTAC and calling it a new approach. If the authors believe their mechanism is novel (not just the receptor or ligand), further evidence to support this claim is needed.

Response. We acknowledge that our degrader operates through the same mechanism as the LYTAC strategy. However, we believe that developing new approaches for constructing degraders is also essential besides the discovery of TPD with new mechanisms. Using the co-assembly concept to construct degraders offers distinct advantages, as demonstrated in this study, which may broaden the applicability of the LYTAC strategy and contribute to the development of novel TPD strategies. In this regard, we assert that our co-assembly approach is innovative in how it constructs the

degrader, and from various perspectives, we would not agree that the co-polymer is equivalent to covalent structures with only slight differences in molecular composition.

Specifically, the major advantages of our co-assembly degrader were summarized as follows:

1) The co-assembly approach simplifies the degrader-construction process by directly mixing the binders modified with assembly-driven motif in a non-covalent manner. This method may avoid laborious synthesis of some complex multivalent binding structures.

2) The “cluster effect” of supramolecular structure imparts multivalent binding capabilities to the degrader. This enables 17A-Lytaca, featuring a component peptide with only 15 negatively charged units “Glu-Ala”, to interaction with the SR-A receptor, which prefers polyanionic ligands. In contrast, a designed SR-A ligand by the Stern group (*Mol. Pharm.* **2020**, *17*, 3794-3812) requires a much larger-sized polymer with 250 negatively charged succinylated Lys units. Our approach demonstrated a distinct efficacy.

3) Beyond SR-A receptor, we extended the LYTACA strategy to degrade PD-L1 on HepG2 through the asialoglycoprotein receptor (ASGPR). It successfully achieved PD-L1 lysosomal degradation with a newly designed PD-L1-Lytaca bearing ASGPR-targeting mono-GalNAc and PD-L1-targeting TPP warheads. This represents a readily accessible approach to amplify the effect of mono-valent structure to achieve effective interaction with receptors that typically demand complex multivalent ligands.

Collectively, our LYTACAs strategy offers a modular and selective platform for constructing LYTAC degraders. This innovative approach may contribute to the development of new protocols for the targeted degradation of both extracellular and membrane proteins. To clarify these points and address the issues raised by the reviewer, more precise statements have been added to our new manuscript in the abstract, intro, and conclusion.

2. Comments on Main Text:

The authors use a combination of inhibitors and treatment conditions to draw the conclusion that SR-A is the receptor which is targeted. However, they also not a contribution of pinocytosis. These experiments are not conclusive in terms of identifying the mechanism of uptake, thus, knockout experiments for SR-A are needed (either CRISPR or siRNA with blot evidence and sequencing). Without these experiments, no mechanistic conclusion can be reached and the work is substantially reduced. The references to multiple cell lines with different levels of SR-A is not evidence of SR-A involvement, merely differential uptake ability of different cell lines.

Response. To further validate the IL-17A uptake mechanism, we used the CRISPR technology to establish a cell line with the knockout of the SR-A receptor (SR-A-KO, new Supplementary Fig. 2). Subsequently, we assessed the IL-17A uptake capability of

the SR-A-KO cell line induced by 17A-Lytaca. The intracellular IL-17A signal exhibited a significant reduction compared to WT cells (new Fig. 1g), providing clear evidence of SR-A involvement in pinocytosis. To illustrate this result, text has been added to the manuscript on page 5 as follows: “To further validate SR-A engagement in IL-17A endocytosis, we established a SR-A receptor knock out cell line (SR-A-KO) with RAW264.7 using the CRISPR technology (Supplementary Fig. 2). The endocytosed IL-17A signal in SR-A-KO cells was significantly lower than in WT RAW264.7 cells, providing conclusive evidence for the involvement of SR-A in IL-17A uptake.”

We acknowledge that the results from this assay are better read as supporting the cell selectivity of 17A-Lytaca, rather than correlating the uptake capability solely with the expression level of SR-A. We have refined the description of the multiple cell lines assay to read “Additionally, different endocytosis efficiency of IL-17A induced by 17A-Lytaca was observed in various cell lines (new Supplementary Fig. S5a), including RAW264.7, U-118MG, DC2.4, and HaCaT, which are known to express different level of SR-A. The western blot result showed that the RAW264.7 cells expressed the highest amount of SR-A among these cell lines (new Supplementary Fig. S5b, 5c), consistent with the IL-17A uptake result.”

Figure. 1g Uptake of IL-17A in wildtype RAW 264.7 cells (WT) or SR-A knockout RAW 264.7 cells (SR-A-KO) incubated with 0.5 $\mu\text{g/ml}$ IL-17A and 50 μM mixed fi-SRAL and fi-17Abp for 6 h.

Supplementary Fig. 2a Western blot of SR-A in wildtype (WT) and SR-A knockout (SR-A-KO) RAW264.7 cells.

Supplementary Fig. 2b Representative flow cytometry results of SR-A level in WT and SR-A-KO RAW264.7 cells.

Supplementary Fig. 5 The IL-17A endocytosis in RAW264.7 cell line is higher than other cell lines. a) IL-17A levels in RAW 264.7, U-118 MG, DC 2.4 and HaCaT cells after a 6 hour-treatment with 0.5 $\mu\text{g/ml}$ IL-17A and 50 μM 17A-Lytaca. Data are presented as the mean \pm SD, and error bars represent the standard deviation of biological replicates ($n = 3$). P values were determined by two-way ANOVA with Sidak's multiple comparisons test. *** $P = 0.0005$, **** $P < 0.0001$, ns, no significant. b) Western blot analysis of SR-A in RAW264.7, U-118MG, DC2.4 and HaCaT cells and c) the corresponding quantified results. Densitometry was used to calculate protein levels, and data ($n = 3$ biologically independent experiments) were normalized to the GAPDH.

3. FITC imaging is used in Fig. 2d, however, FITC is well-known to be quenched in acidic environments, and LysoTracker is not a specific marker for lysosomes. LAMP1 imaging is more accurate for claiming colocalization with lysosomes and should be performed here.

Response. We have conducted the co-localization assay employing. Confocal images displayed the co-localization of IL-17A, peptides and lysosome LAMP1 (new Figure. 1h), indicating the endocytosed IL-17A induced by the peptide mixture was transported to lysosomes for degradation. The corresponding text has been revised accordingly.

Figure. 1h Confocal microscopy images depict RAW264.7 cells treated for 6 h with 0.5 $\mu\text{g/ml}$ RED-tris-NTA-labeled IL-17A and 50 μM mixed fi-SRAL and fi-17Abp, showing co-localization of IL-17A (violet), lysosome (red), as well as peptides (green). Scale bar, 10 μm . The intensity profiles of peptides mixture, IL-17A and LAMP1 along the white line are plotted in the right panels.

4. The authors have left out key control experiments and comparisons for their in vivo data. A major component which needs to be addressed is treatment with the SRAL

peptide alone. Without this control, it is not clear if a general macrophage activation strategy with increased inflammation underlies their observed effect with the combination of SRAL and 17Abp. The authors should directly compare the 17A-LYTACA to 17-Abp for clearance of IL17-A in Fig. 4g and 4h. The effect size difference is small, if significant at all, and thus is unclear if clearance of IL17-A with a Lytaca is actually occurring in vivo or underlies the observed pathology. Indeed, without the SRAL peptide treatment alone, the cytokine mRNA profiling is preliminary at best. P values were calculated using a t-test, but the more statistically relevant method would be with an ANOVA across all groups. Overall the in vivo data is not convincing, especially without controls or adequate statistically comparisons.

Response. Thanks to the reviewer’s suggestion, we incorporated a SRAL peptide control group and repeated the in vivo experiment. The results revealed that SRAL alone exhibited a similar performance to 17Abp alone, both of which failed to alleviate psoriasis manifestation (new Fig. 4). Therefore, we attribute the observed reduction in psoriasis severity to the effects induced by 17A-Lytaca rather than a general macrophage activation with increased inflammation. To clarify this point, the results have been included in the revised manuscript on page 9 as follows:

“Accordingly, we employed an IMQ-induced psoriasis-like mouse model³⁶, where BALB/c mice were treated with IMQ cream on their shaved back skin for seven consecutive days, followed by intraperitoneal injection of 17A-Lytaca, methotrexate (MTX), fm-17Abp alone, fm-SRAL alone, or saline as control (Figure 4a).

As anticipated, the results showed that 17A-Lytaca treatment successfully reduced the severity of psoriasis manifestation compared to the positive group. This was evidenced by factors such as the body weight loss (Fig 4b), splenomegaly (Figure 4c), both clinical and pathological characteristics (Figure 4d), acanthosis (Figure 4e), and disease severity, including a reduction in skin thickness, erythema, and scaling (Figure 4f, 4g, Supplementary Fig. 7). Interestingly, treatment with fm-17Abp or fm-SRAL alone showed no significant difference compared to the IMQ model group, highlighting the superiority of our co-assembly strategy over inhibitor treatment. In addition, the 17A-Lytaca treatment demonstrated better performance in slowing down weight loss (Fig. 4b) and reducing the PASI score (Fig. 4f) than the MTX group, underscoring the superior biocompatibility and safety of our co-assembly system. Notably, the 17A-Lytaca treatment did not show any obvious toxicity to the organs of mice (Supplementary Fig. 8).

To further validate the IL-17A-targeting capability of 17A-Lytaca in vivo, we measured the IL-17A levels in mouse dorsal skin. Impressively, the 17A-Lytaca treatment resulted in a significant reduction of IL-17A in the skin lesion (Fig. 4h). In contrast, the SRAL- or 17Abp-treated group exhibited no obvious effects on changing IL-17A levels compared to the IMQ model group. Remarkably, treatment of 17A-Lytaca markedly decreased the mRNA expression of IL-17A and its downstream psoriasis-related inflammatory cytokines in the skin lesion, including IL-6, CCL-20, and IL-22, compared with the positive group. Conversely, the 17Abp and SRAL alone showed no difference from the positive group (Fig. 4i).”

In the *in vivo* IL-17A clearance characterization assay, we observed low serum IL-17A levels, consistent with previous reports (*J. Invest. Dermatol.* **2019**, *139*, 638-647) and showing only minor differences between healthy and psoriatic mice. Consequently, using this indicator to illustrate the impact of our intervention on psoriatic symptoms proved challenging. However, significant changes in IL-17A levels in dorsal skin were observed, clearly demonstrating *in vivo* IL-17A decline induced by Lytaca and MTX compared to the saline group (new Fig. 4h). Therefore, in our new *in vivo* experiment, we chose to utilize the change in IL-17A levels in dorsal skin to illustrate the impact of our intervention on IL-17A clearance.

Moreover, we thank the reviewer for pointing out the issue regarding the calculation of P values. We have re-assessed the statistical significance using the One/Two-Way ANOVA method across all groups.

new Fig. 4

Figure 4. LYTACAs attenuate skin inflammation in psoriasis mouse model via IL-17A clearance. a) Experimental timeline. For 7 consecutive days, Saline, fm-17Abp

(15 mg/kg/day), fm-SRAL (22 mg/kg/day), 17A-Lytaca (fm-17Abp and fm-SRAL mixture was given 15 and 22 mg/kg/day after preincubated for 12 h) or MTX (1 mg/kg/twice day) were intraperitoneally administered into BALB/c mice after the topical application of 5% IMQ cream. **b**) Daily body weight changes. **c**) Splenomegaly degree. **d**) Representative clinical manifestations and H&E staining of the back skin on day 7, scale bar: 100 μ m. **e**) Analysis of acanthosis change on day 7. **f, g**) PASI score reflecting the severity of skin erythema, scaling and thickening for 7 days. **h**) IL-17A protein level in skin lesion on day 7. **i**) Relative mRNA expression of IL-17A, IL-6, CCL-20 and IL-22 in skin lesion. **j, k**) Representative *in vivo* and organ fluorescence images (n = 3) of psoriasis mice at 6 h after intraperitoneal treatment with Cy-labeled 17A-Lytaca, comprising Cy5 labeled fm-17Abp (5 mg/kg) and Cy7 labeled fm-SRAL (7.3 mg/kg). **l, m**) Representative *in vivo* and skin fluorescence images of psoriasis mice or healthy mice at 6 h after intraperitoneal treatment with Cy-labeled 17A-Lytaca, comprising Cy5 labeled fm-17Abp (5 mg/kg) and Cy7 labeled fm-SRAL (7.3 mg/kg). For **b, c** and **e-i**, Data are presented as the mean \pm SD, and error bars represent the standard deviation of biological replicates (n = 5). P values were determined by one-way ANOVA with Tukey's multiple comparisons test (**c, e, h, i**) or two-way ANOVA with Sidak's multiple comparisons test (**b, f, g**). **b** ****P < 0.0001; **c** **P = 0.0013, ****P < 0.0001; **e** ****P < 0.0001; **f** ****P < 0.0001; **g** ****P < 0.0001; **h** **P = 0.0027, ****P < 0.0001; **i** **P = 0.0092, ****P < 0.0001. ns, no significance.

5. Further, adequate PK experiments must also be presented, such as clearance time of the relevant peptides from serum, and if the conjugate is capable of reaching the relevant location given the skin depletion data. Additionally, what evidence exists that the peptide complexes going to remain *in vivo*?

Response. To study the serum clearance kinetics of the co-polymer, Cy-labeled 17A-Lytaca, consisting of Cy3-labeled fm-SRAL and Cy5-labeled fm-17Abp, was intraperitoneally injected into healthy mice. Blood samples were collected at various time points (0, 1, 2, 3, 4, 8, 12, 24 h), and the fluorescence intensity of the serum was measured using the *in vivo* imaging system (IVIS). The fluorescent signals of both peptides were detectable by IVIS within the initial 8 hours. Subsequently, fm-SRAL signal disappeared, while fm-17Abp persisted until 24 hours (new Supplementary Fig. 9).

Meanwhile, we investigated the *in vivo* distribution and clearance of the co-polymer in psoriasis mice with 17A-Lytaca, comprising Cy5-labeled fm-17Abp and Cy7-labeled fm-SRAL. The Cy-labeled 17A-Lytaca was administered to both psoriasis mice and healthy mice via intraperitoneal injection. Fluorescence images in both Cy5 and Cy7 channels were captured using the IVIS spectrum at 4, 8, 12, 24 and 48 h post-injection. The result showed both peptides remained detectable with co-localized signal for up to 48 hours in psoriasis mice (new Supplementary Fig. 10), providing evidence for the persistence of peptide complexes *in vivo*.

Furthermore, for biodistribution assessment, mice were sacrificed at 6 h post-injection, and skins along with major organs (heart, liver, spleen, lung and kidney) were removed for ex vivo fluorescence imaging. Both peptides were able to reach the skin lesion of psoriasis mouse, besides accumulating in the liver and kidney (new Fig. 4j, k). In contrast, 17A-Lytaca was not observed on the dorsal skin of healthy mouse. These results underscore the tissue selectivity of our approach (new Fig. 4l, m).

To clearly illustrated these results, we added the text to the manuscript on page 9 as follows: “The clearance kinetics of the peptide components within 17A-Lytaca were also investigated by fluorescently labeling fm-SRAL with Cy3 and fm-17Abp with Cy5. Both peptides' fluorescent signals were detectable by in vivo imaging system (IVIS) within the initial 8 hours. Subsequently, fm-SRAL disappeared, while fm-17Abp persisted until 24 hours in healthy mice (Supplementary Fig. 9). In psoriasis mice, both peptides remained detectable until 48 hours (Supplementary Fig 10). The results also indicated that the co-polymer could reach and persist in the skin lesion and spleen in IMQ-induced psoriasis mice, while the relevant organs of healthy mice showed no peptide signal (Fig. 4j, k, l, m).”

new Supplementary Fig. 9

new Supplementary Fig. 10

Supplementary Fig. 9 Serum clearance kinetics assay. The experiment was conducted in healthy mice with Cy-labeled 17A-Lytaca, comprising Cy5-labeled fm-17Abp (15 mg/kg) and Cy3-labeled fm-SRAL (22 mg/kg). **a)** Time-dependent fluorescence images of serum captured in Cy5 channel. **b)** Quantitative Cy5 fluorescence intensity of 17A-Lytaca. **c)** Time-dependent fluorescence images of serum

captured in Cy3 channel. **d)** Quantitative Cy3 fluorescence intensity of 17A-Lytaca. Data are presented as the mean \pm SD, and error bars represent the standard deviation of biological replicates (n = 3).

Supplementary Fig. 10 17A-Lytaca diffused in the skin lesion and exhibited prolonged in vivo retention in psoriasis mice. Time-dependent *in vivo* fluorescence images of psoriasis mice after intraperitoneal treatment with Cy-labeled 17A-Lytaca, comprising Cy5 labeled fm-17Abp (5 mg/kg) and Cy7 labeled fm-SRAL (7.3 mg/kg)

Figure. 4j, k Representative *in vivo* and organ fluorescence images (n = 3) of psoriasis mice at 6 h after intraperitoneal treatment with Cy-labeled 17A-Lytaca, comprising Cy5 labeled fm-17Abp (5 mg/kg) and Cy7 labeled fm-SRAL (7.3 mg/kg).

Figure. 4l, m Representative *in vivo* and skin fluorescence images of psoriasis mice or healthy mice at 6 h after intraperitoneal treatment with Cy-labeled 17A-Lytaca, comprising Cy5 labeled fm-17Abp (5 mg/kg) and Cy7 labeled fm-SRAL (7.3 mg/kg).

6. For the PD-L1 degradation experiments in Fig. 5, the authors must also include mechanistic experiments which demonstrate their peptides form similar complexes, and are utilizing the SR-A receptor through knock-out experiments. Further, quantification of the Western blot degradation results in Fig. 5d in biological triplicate in bar graph form is the standard of the field. Additionally, evidence that SR-A is recycling, as their diagram in Fig. 5a demonstrates should be presented in the form of Western blots of degradation experiments.

Response. We studied the assembly behavior of our degrader using TEM. The results revealed that nf-PD-L1-Lytaca formed spherical nanoparticles (new Fig. 5f), while either nf-TPP or nf-SRAL alone exhibited fibrous structures (new Supplementary Fig. 13). Moreover, PD-L1 was successfully degraded by PD-L1-Lytaca, whereas it remained intact in SR-A-KO cell lines, indicating the involvement of SR-A receptor (new Fig. 5c). Time-dependent PD-L1 degradation induced by PD-L1-Lytaca was observed, coinciding with the decline and subsequent recycling of the SR-A receptor (new Fig. 5i). Furthermore, PD-L1 degradation exhibited a concentration-dependent response to our co-polymer (new Fig. 5h), with the western blot result presented in bar graph form for biological triplicates (new Supplementary Fig 13a).

These results were described in our revised manuscript on page 11-12 as follows: “Concentration-dependent responses were observed in PD-L1 degradation in RAW264.7 cells treated with PD-L1-LYTACA, peaking at 100 μ M. In contrast, neither fm-TPP nor fm-SRAL alone showed any discernible impact on degradation at a concentration of 50 μ M (Fig. 5b). Importantly, the absence of PD-L1 degradation in SR-A-KO cells (Fig 5c) conclusively demonstrates the involvement of SR-A in this TPD process.” and

“TEM showed that the nf-modified PD-L1-Lytaca formed spherical nanoparticles (Fig. 5f), while nf-TPP and nf-SRAL alone formed fibers (Supplementary Fig. 12).” and

“The reduction of PD-L1 induced by nf-modified Lytaca proved to be both concentration and treatment time-dependent, reaching a peak degradation exceeding 70% in RAW264.7 cells (Fig 5 h, 5i, Supplementary Fig. 13a). Notably, during this TPD process, the SR-A level exhibited a initial decrease followed by recovery over time (Fig. 5i), suggesting that Lytaca did not result in the depletion of SR-A. Collectively, these findings underscore the potential of LYTACAs to induce the degradation of cell membrane proteins in macrophages and tumor cells through the involvement of SR-A.”

Figure. 5f Representative TEM image of nf-PD-L1-Lytaca at a concentration of 400 μM in H₂O. Scale bar, 50 nm.

Supplementary Fig. 12 nf-TPP or nf-SRAL alone showed structure of fiber. Representative TEM images for **a)** nf-TPP and **b)** nf-SRAL in H₂O at a concentration of 400 μ M. Scale bar, 200 nm.

Figure. 5c Western blot of PD-L1 in wildtype (WT) or (SR-A-KO) RAW264.7 cells treated with 50 μ M fm-PD-L1-Lytaca for 12 h.

Figure. 5i Western blot of PD-L1 and SR-A in RAW264.7 cells treated with 50 μ M nf-PD-L1-Lytaca for 2, 4, 8, 12, 24 h.

Figure. 5h Dose escalation experiment for PD-L1 degradation in RAW264.7 cells treated with 10, 25, 50 and 100 μ M nf-PD-L1-Lytaca or 50 μ M nf-TPP/nf-SRAL for 12 h.

Supplementary Fig. 13a Western blot analysis of membrane PD-L1 in RAW264.7 cells treated with 10, 25, 50 and 100 μ M nf-PD-L1-Lytaca, 50 μ M nf-TPP or nf-SRAL for 12 h. Data are presented as the mean \pm SD, and error bars represent the standard deviation of biological replicates (n = 3).

7. Comments on Discussion: Overall, not enough evidence is presented to suggest this is a general approach for peptide assembly. While the self-assembled structures are characterized in vitro, it is unclear exactly what drives their assembly. For this to be a new class of extracellular degraders, there must be some logic for assembly of any target binder into these complexes. However, the lack of characterization of the PD-L1 Lytaca, and the unclear explanation for self-assembly of the IL-17 molecule makes it difficult to judge if these complexes are the same for different targets. The comment "In addition, when compared to covalent chimeras that demand a strenuous optimization process, this co-assembly degradation system is relatively straightforward to prepare using diverse peptide binders" does not make sense. There is no evidence that strenuous linker optimization efforts must be applied to extracellular binders, and the rate limiting step for any degrader is the necessity of generating a peptide binder in the first place. Overall, significantly more work must be done to rigorously demonstrate the nature of the complexes, their generality, and their specificity.

Response. Thanks for the insightful opinions from Reviewer #2. **Regarding the logic for assembling target binders into the complexes**, we attribute the assembly-driven force to the N-terminal aromatic motif, such as Fmoc, which possesses the ability to self-assemble through hydrophobic interactions and π - π stacking. Besides Fmoc, structures like naphthalene (Nap), and naphthalene-Phe-Phe (Nap-FF) are widely studied and applied in producing supramolecular peptide assemblies (*ACS Appl. Bio. Mater.* **2023**, *6*, 384-409; *Adv. Mater.* **2006**, *18*, 611-614; *Adv. Mater.* **2006**, *18*, 1365-1370). Therefore, incorporating such structures to peptide binders of target proteins could be a general approach.

To present a clearer illustration on the peptide assembly, we conducted additional characterizations of our LYTACAs. Firstly, a surfactant competitive assay was

performed using sodium dodecylbenzenesulfonate (SDBS), a surfactant containing a benzene structure. The TEM image revealed that the addition of SDBS perturbed the original morphology and size of 17A-Lytaca (new Supplement Fig. 4c). We reason that the benzene moiety of SDBS may interact with the aromatic rings of the Fmoc on the peptide N-terminus, thereby disrupting the co-assembly process of the two peptides. This also supports the assembly-driven capability of the Fmoc motif. Additionally, the critical aggregation concentration of our complexes was measured using DLS and calculated as 11.75 μM (new Supplementary Fig. 4d). The time dependent stability of the co-assembled supramolecular complex formed by the two peptides was also assessed by DLS at both 37 °C and 4 °C, showing the maintained size and monodispersity over a period of 12 h (new Fig 2e, new Supplementary Fig. 4e). Moreover, the assembly behavior of PD-L1-Lytaca were characterized, demonstrating the formation of similar spherical nanoparticle complex (new Fig 5f) as the 17A-Lytaca in contrast to fibrous structures of their component peptides alone (new Supplementary Fig.12).

To further demonstrate the generality of our approach, we employed a Nap-FF structure (new Fig. 5e), known for its robust assembly-driving capability, to replace the N-terminal Fmoc motif, generating the nf-PD-L1-Lytaca. Western blot showed improved efficacy with exceeding 70% degradation of PD-L1 in RAW264.7 cells (new Fig. 5g, h). Furthermore, our co-polymers effectively induced PD-L1 degradation on A549 and HepG2 cancer cell lines with the SR-A receptor, using both fm-PD-L1-Lytaca (new Fig. 5d, new Supplementary Fig 11c) and nf-PD-L1-Lytaca (new Supplementary Fig 13b, new Supplementary Fig 13c), highlighting the universality of this technology.

The need for rigorous linker optimization efforts for extracellular binders is underscored by existing evidence. Researchers had reported that the length of the linker in LYTAC degraders is crucial for target degradation, necessitating thorough screening (*ACS Chem. Biol.* **2023**, *19*, 1611-1623). For example, Fang et al. demonstrated that variations in linker length could lead to a significant difference in PD-L1 degradation efficiency, spanning from 16% to 63% (*J. Am. Chem. Soc.* **2022**, *144*, 21831-21836). This observation is consistent with findings in PROTAC cases (*Nat. Rev. Clin. Oncol.* **2023**, *20*, 265-278; *Eur. J. Med. Chem.* **2020**, *201*, 112451).

new Supplementary Fig. 4c

new Supplementary Fig. 4d

new Fig. 2e

new Supplementary Fig. 4e

new Fig. 5f

new Supplementary Fig. 12

new Fig. 5e

new Fig. 5g

new Fig. 5h

new Fig. 5d

new Supplementary Fig. 11c

new Supplementary Fig. 13b

new Supplementary Fig. 13c

Supplementary Fig. 4c Representative TEM images for Mixed fm-SRAL and fm-17Abp in H₂O or sodium dodecylbenzene sulfonate (SDBS, 800 μM) buffer at a concentration of 400 μM. Scale bar, 100 nm.

Supplementary Fig. 4d The critical aggregation concentration (CAC) of the co-assembled complex comprising fm-SRAL and fm-17Abp (molar ratio 1:1). Peptides were incubated at 4 °C for 12 h and then detected by dynamic light scattering (DLS) for CAC calculation.

Figure. 2e Time-dependent changes in the hydrodynamic size (red) and polydispersity index (PDI, grey) of the co-assembled supramolecular complex formed by fm-17Abp and fm-SRAL at 37 °C.

Supplementary Fig. 4e Time-dependent changes in the hydrodynamic size (red) and polydispersity index (PDI, grey) of the co-assembled supramolecular complex formed by fm-17Abp and fm-SRAL at 4°C.

Figure. 5f Representative TEM image of nf-PD-L1-Lytaca at a concentration of 400 μM in H₂O. Scale bar, 50 nm.

Supplementary Fig. 12 nf-TPP or nf-SRAL alone showed structure of fiber. Representative TEM images for **a)** nf-TPP and **b)** nf-SRAL in H₂O at a concentration of 400 μM. Scale bar, 200 nm.

Figure. 5e Chemical structure of N-terminal modification group Fmoc and Nap-FF.

Figure. 5g Western blot of PD-L1 in RAW264.7 cells treated with 50 μM fm-/nf-PD-L1-Lytaca, fm-/nf-TPP or fm-/nf-SRAL for 12 h.

Figure. 5h Dose escalation experiment for PD-L1 degradation in RAW264.7 cells treated with 10, 25, 50 and 100 μM nf-PD-L1-Lytaca or 50 μM nf-TPP/nf-SRAL for 12 h.

Figure. 5d Dose escalation experiment for PD-L1 degradation in A549 cells treated with 10, 25, 50 and 100 μM fm-PD-L1-Lytaca or 50 μM fm-TPP/fm-SRAL for 12 h

Supplementary Fig. 11c Western blot of PD-L1 in HepG2 cells treated with 50 μM fm-TPP, fm-SRAL or fm-PD-L1-Lytaca for 12 h.

Supplementary Fig. 13b Western blot of PD-L1 in HepG2 cells treated with 50 μM nf-TPP, nf-SRAL or nf-PD-L1-Lytaca for 12 h.

Supplementary Fig. 13c Western blot of PD-L1 in A549 cells treated with 50 μM nf-TPP, nf-SRAL or nf-PD-L1-Lytaca for 12 h.

Reviewer #3.

1. The manuscript by Wang et al. introduces a methodology for the targeted degradation of extracellular proteins, building upon the principles of the LYTAC approach used to target surface receptors. However, this new approach, called lysosome-targeting co-assemblies (LYTACA), has notable differences. Firstly, it utilizes class A scavenger receptors (SR-A) to direct cargo toward lysosomal degradation. Secondly, it employs two separate peptides that can assemble together.

The study provides evidence that Lytaca targeting IL-17A promotes the internalization of IL-17A by the RAW264.7 macrophage cell line. Furthermore, the authors show that IL-17A-Lytaca can partially reduce disease progression in an imiquimod-induced psoriasis mouse model. Additionally, the authors demonstrate the ability of Lytaca particles that binds PD-L1 to target and downregulate surface PD-L1 on RAW264.7 macrophages.

However, a key conceptual issue with this proposed approach arises from the rapid internalization of the SR-A-targeting peptide (as depicted in Figure 1). Based on this observation, it is likely that LYTACA particles would be swiftly cleared from circulation, even if they fail to bind the presumed cargo proteins like IL-17A. This would limit the system's ability to remove IL-17A that is continuously released at the site of inflammation. The only conceivable solution would be to administer large amounts of IL-17A-Lytaca on a regular basis – presumably daily, as shown in the in vivo experiment (Figure 4). This seems very disadvantageous for use in clinical settings.

In contrast, targeting surface proteins like PD-L1 may offer certain advantages since the internalized LYTACA particles would be more likely to associate with the target protein. Nevertheless, even under most optimal conditions using the RAW264.7 cell line, adding PD-L1-Lytaca leads to only a 30% decrease in surface PD-L1, as shown in Figure 5. This level of degradation is unlikely to be relevant in cancer settings, particularly considering that PD-L1 would presumably only be degraded on SR-A-expressing cells. Consequently, the presented LYTACA method does not appear to be very promising at the current stage.

Response. We appreciate the reviewer's comments and suggestions. We thought this key conceptual issue might be the intrinsic problem of LYTAC and related strategies for degrading extracellular proteins. One way circumventing this issue might be optimizing the degrader to achieve faster and more efficient binding to the targeted protein. This optimization could be accomplished through various approaches, and our co-assembly strategy offers one relatively convenient option. While the primary focus of this article is to establish the feasibility of applying the co-assembly strategy in constructing degraders for TDP, we acknowledge the need for further optimization for clinical applications in our future research.

As illustrated in the case of PD-L1 degradation, optimizing the PD-L1-Lytaca structure by replacing the Fmoc motif with Nap-FF (new Fig. 5e) resulted in over 70% degradation of PD-L1 in RAW264.7 cells (new Fig. 5g), highlighting its promising

application in cancer settings. Furthermore, PD-L1-Lytaca effectively induced lysosomal PD-L1 degradation in cancer cell lines expressing SR-A, such as HepG2 (new Supplementary Fig 13b) and A549 (new Supplementary Fig 13c). The modularity of this LYTACA system also allows for the exploration of other receptors, such as ASGPR demonstrated in this study, to develop new degraders more suited for specific applications.

To highlight this point and present the experiment results, text have been incorporated into the manuscript on Page 12 as follows: “To enhance degradation efficiency, we sought to optimize the structures of the assembling motifs. Considering the influential roles of the hydrophobic N-termini in increasing endocytosis (Figure 2c), we prepared an improved PD-L1-Lytaca by replacing Fmoc with Nap-FF (abbreviated as nf), a structure with robust assembly-driving capability^{42, 43, 44} owing to enhanced π - π stacking (Fig. 5e).” and “Remarkably, compared with fm-PD-L1-Lytaca, nf-PD-L1-Lytaca exhibited significantly enhanced degradation efficiency in macrophages (Fig. 5g) and cancer cells (Supplementary Fig. 13b, c).”

Figure. 5e Chemical structure of N-terminal modification group Fmoc and Nap-FF.

Figure. 5g Western blot of PD-L1 in RAW264.7 cells treated with 50 μ M fm-/nf-PD-L1-Lytaca, fm-/nf-TPP or fm-/nf-SRAL for 12 h.

Supplementary Fig. 13b Western blot of PD-L1 in HepG2 cells treated with 50 μ M nf-TPP, nf-SRAL or nf-PD-L1-Lytaca for 12 h.

Supplementary Fig. 13c Western blot of PD-L1 in A549 cells treated with 50 μ M nf-TPP, nf-SRAL or nf-PD-L1-Lytaca for 12 h. Densitometry was used to calculate protein levels, and data were normalized to the PBS control.

Several key points to be addressed:

2. The authors demonstrate that IL-17A-Lytaca particles can promote the internalization of IL-17A. However, it is important to determine the extent to which this internalization depletes the provided IL-17A from the medium to evaluate the proposed approach's efficiency.

Response. We assessed the levels of both IL-17A in the medium over time following the treatment of RAW 264.7 cells with 17A-Lytaca and IL-17A for the specified duration. The western blot results indicated that the level of IL-17A in the medium did not exhibit observable changes (Fig. R1a, b), suggesting an overdose of the provided protein.

Nevertheless, our 17A-Lytaca demonstrated a rapid uptake of IL-17A within 6 hours prior to its depletion (new Fig. 3d). Furthermore, in clinical settings, the reported concentration of IL-17A in psoriatic lesions is 0.01 ng/ml (*Sci. Rep.* **2016**, *6*, 26071; *J. Immunol.* **2006**, *177*, 36–39), significantly lower than the concentration (0.5 μ g/ml) we used in our in vitro uptake assay. Therefore, the efficiency of LYTACA may be sufficient to achieve the degradation of IL-17A at levels relevant to those observed in patients.

Figure. R1a Western blot of IL-17A in culture medium in a time-course experiment for IL-17A uptake, in which the RAW264.7 cells were treated with 0.5 μ g/ml IL-17A and 50 μ M mixed fi-SRAL and fi-17Abp for 6 h. The anti-His tag antibody was applied in the WB assay to indicate the trace of IL-17A bearing His tag.

Figure. R1b The corresponding quantified data of the western blot result in **Figure R1a**.

Figure. 3d Time-course experiment for IL-17A uptake in RAW264.7 cells incubated with 0.5 μ g/ml IL-17A and 50 μ M mixed fi-SRAL and fi-17Abp for 1, 3, 6, 12, 24 and 48 h.

3. It is possible that upon binding to SR-A, the Lytaca particles can directly activate a signaling response. Therefore, it is necessary to investigate whether the treatment of cells with LYTACA leads to cellular activation.

Response. To explore whether 17A-Lytaca particles trigger SR-A-mediated activation of signaling responses, RAW264.7 cells were with 50 μ M 17A-Lytaca, fm-17Abp, or fm-SRAL alone for 6 h. Western blot analysis was conducted to evaluate the phosphorylation levels of ERK, a downstream signaling molecule of SR-A (*Nat. Commun.* **2023**, *14*, 4895; *J. Biol. Chem.* **2001**, *276*, 28719-28730; *J. Biol. Chem.* 2011, *286*, 8231-8239). The western blot results showed that treatment with 17A-Lytaca increased the phosphorylation level of ERK, indicating cellular activation induced by 17A-Lytaca (Fig. R4). This finding aligns with previous studies (*Nat. Commun.* **2023**, *14*, 4895; *Respir. Res.* **2008**, *9*, 59) where the ERK signaling pathway was implicated in SR-A-mediated uptake, providing further evidence supporting the engagement of SR-A in our system. However, since the primary focus of this study is to establish the concept of applying the co-assembly strategy in constructing degraders for targeted protein degradation, further in-depth investigations into the mechanistic aspects will be pursued in subsequent studies.

Figure. R4a Schematic illustration of the cellular activation of ERK signaling pathway by SR-A-mediated uptake of 17A-Lytaca.

Figure. R4b Western blot of ERK and p-ERK in RAW264.7 cells treated with 50 μ M 17A-Lytaca, fm-17Abp or fm-SRAL alone for 6 h.

4. Related to the previous point, the psoriasis experiment in Figure 4 is missing control to rule out the possibility that IL-17A-Lytaca particles trigger SR-A-mediated activation of signaling responses, which might be independent of IL-17 depletion. The author should prepare Lytaca particles with a control peptide that does not bind to IL-17A and confirm that the administration of this control-Lytaca does not result in the observed effects in the *in vivo* experiment.

Response. Thanks to the helpful suggestions from reviewer #3, we re-conducted the *in vivo* experiments with an additional control-Lytaca co-assembly. This co-assembly

contains a newly designed control peptide (Fig. R5a), which is capable of forming co-assembled particles with SRAL (Fig. R5b) but exhibiting no binding to IL-17A, as confirmed by SPR experiments (Fig. R5c). The results showed that control-Lytaca exhibited the similar performance with 17Abp alone which could not alleviate the psoriasis manifestation (Fig. R5d, e, f, g, h, I, j), demonstrating the requirement of 17A-Lytaca particles for the therapeutic efficacy.

Figure. R5a Sequence of control peptide.

Figure. R5b Representative TEM images for control peptide and mixture of control peptide+ fm-SRAL at a concentration of 400 μ M. Scale bar, 100 nm.

Figure. R5c SPR curve of control peptide at the indicated concentration binding to mouse IL-17A immobilized on the CM5 chip surface.

Figure. R5d Daily body weight changes.

Figure. R5e Splenomegaly degree.

Figure. R5f PASI score reflecting the severity of skin erythema, scaling and thickening for 7 days.

Figure. R5g Representative clinical manifestations and H&E staining of the back skin on day 7, scale bar: 100 μ m.

Figure. R5h Analysis of acanthosis change on day 7

Figure. R5i Relative mRNA expression of IL-17A, IL-6, CCL-20 and IL-22 in skin lesion.

Figure. R5j IL-17A protein level in skin lesion on day 7.

5. Minor points:

1). In several figures, such as Figure 3e, or 5b-c, the y-axis is segmented into two parts without an apparent reason. It seems that the purpose is solely to visually emphasize the differences. It would be more appropriate to present these graphs with a non-segmented y-axis.

Response. Revisions have been made.

2). The effect of Lytaca on psoriasis progression, as shown in Figures 4b and 4c, should be accompanied by statistical analysis. The observed effect seems to be weak, and conducting statistical tests will provide a more robust evaluation.

Response. Revisions have been made.

3). The results presented in Figure 4e, demonstrating an increase in skin thickness, should be quantified and displayed in a separate graph.

Response. We have quantified the acanthosis thickness of all groups and displayed the result in a separated graph (new Fig. 4e) in the revised manuscript on page 10.

Figure. 4e Analysis of acanthosis change on day 7.

4). The description of the statistical analysis is lacking. It is unclear why the authors used one-way ANOVA to compare two samples in Figures 1, 2, 3, and 5, as this test is typically employed for comparing more than two groups. In contrast, a t-test is used in Figure 4. The methods section should provide an explanation for the choice of specific statistical tests. Additionally, information about how the normality of the data was determined, especially if using tests that assume a normal distribution, should be included.

Response. We have re-assessed the statistical significance using One-Way ANOVA with Tukey's multiple comparisons test or two-way ANOVA with Sidak's multiple comparisons test for all data. Normality of data sets was assumed for ANOVA and was tested by Shapiro-Wilk tests.

5). Some of the statistical analyses yield surprising results. For example, in Figure 3f, there is a very clear distinction between the internalization of IL-17 by HaCaT cells in the presence or absence of IL-17A-Lytaca. However, the authors state that it is not statistically significant. This raises questions about the appropriateness of the statistical tests used.

Response. In the statistical analyses of the IL-17 uptake assay in multiple cell lines, a two-way ANOVA method with Sidak's multiple comparisons test was employed (*P < 0.05, **P < 0.01, ***P < 0.001, ****P < 0.0001). This analysis was conducted to compare the mean IL-17A uptake between each cell line with the mean of other cell lines in the same row. Notably, the endocytosis of IL-17A by RAW264.7 was significantly higher than the other three cell lines, making their internalization of IL-17A, in the presence or absence of IL-17A-Lytaca, statistically non-significant in contrast. However, when analyzing the same cell lines in the presence or absence of IL-17A-Lytaca using the t-test method, significant differences were revealed. Given that the main point of this assay is to compare the IL-17A uptake capability across these different cell lines, the two-way ANOVA method was deemed more suitable for this purpose.

new Supplementary Fig. 5a

Supplementary Fig. 5a IL-17A levels in RAW 264.7, U-118 MG, DC 2.4 and HaCaT cells after a 6 hour-treatment with 0.5 $\mu\text{g/ml}$ IL-17A and 50 μM 17A-Lytaca. Data are presented as the mean \pm SD, and error bars represent the standard deviation of biological replicates ($n = 3$). P values were determined by two-way ANOVA with Sidak's multiple comparisons test. *** $P=0.0005$, **** $P < 0.0001$, ns, no significance.

Reviewer #4.

In this paper, Wang et al developed self-assembled structure of two binding proteins. They co-assembled SR-A peptide ligand and IL-17A binding domain. The resulting assembled structure efficiently removed extracellular IL-17A by lysosomal degradation in both cell lines and mouse models. In addition, the authors applied this system to remove PD-L1 receptor on cell surface, not in extracellular environment. I think that this study is meaningful, but premature for publication in my opinion because key data is missing. I recommend the publication after major revision as below.

1. What is the main reason of the increased effect of self-assembled structure? It is the key point and should be determined by experimental data and sufficient discussion.
2. In addition, why did the covalent fi-SRAL-17Abp chimera show no increase of IL-17A uptake in cells?

Response. We appreciate the reviewer's feedback and valuable suggestions. Certain receptors capable of mediating endocytosis, including SR-A and ASGPR, are multivalent, requiring high-affinity ligand binding through multivalent interactions (*Nat. Chem. Biol.* **2021**, *17*, 937-946; *ACS Cent. Sci.* **2021**, *7*, 499-506; *Angew. Chem. Int. Ed. Engl.* **2023**, *62*, e202300694.). For example, a designed SR-A ligand by the Stern group requires a large-sized polymer with 250 negatively charged succinylated Lys units to ensure effective binding (*Mol. Pharm.* **2020**, *17*, 3794-3812). Therefore, the covalent construct, fi-SRAL-17Abp bearing limited anionic content (15 negatively charged units), may not be sufficient enough to bind SR-A. In contrast, the “cluster effect” of supramolecular structure imparts strong multivalent binding capabilities to the degrader, which enables 17A-Lytaca to interact with SR-A and facilitates the endocytosis of IL-17A.

In the case of ASGPR, which requires multivalent ligand binding (e.g., tri-GalNAc), we also found that the LYTACA system could amplify the effect from a mono-valent GalNAc ligand to result effective binding. This is evidenced by successful degradation of PD-L1 in HepG2 cells using a PD-L1-Lytaca bearing ASGPR-targeting mono-GalNAc and PD-L1-targeting TPP warheads (new Fig. 5j, 5k). This result further demonstrated the universality and versatility of our approach.

Figure. 5j Schematic illustration of ASGPR binding with Tri-GalNAc or fm-PD-L1-Lytaca consist of TPP ligand and mono-GalNAc ligand.

new Figure. 5k Western blot of PD-L1 in HepG2 cells treated with 50 μ M fm-TPP, ASGPRL or mixed fm-TPP and ASGPRL (fm-TPP+ASGPRL) for 12 h.

3. In studies of self-assembled structure, its time-dependent stability and critical concentration for self-assembly are important information and should be provided.

Response. We have assessed the time-dependent stability and critical concentration for 17A-Lytaca. The co-assembled structure maintained stable within 12 hours at both 37 $^{\circ}$ C (Fig. 2e) and 4 $^{\circ}$ C (Supplementary Fig. 4e). The critical concentration for co-assembly was determined as 11.75 μ M (Supplementary Fig. 4d). These results have been incorporated into the revised manuscript on the page 8 as follows: “Assessed by DLS²⁹, the critical aggregation concentration for the co-assembly was determined as 11.75 μ M (Supplementary Fig. 4d), and this supramolecular complex maintained the size and monodispersity over a 12-hour period at both 37 $^{\circ}$ C and 4 $^{\circ}$ C. (Fig 2e, Supplementary Fig. 4e).”

Figure. 2e Time-dependent changes in the hydrodynamic size (red) and polydispersity index (PDI, grey) of the co-assembled supramolecular complex formed by fm-17Abp and fm-SRAL at 37 $^{\circ}$ C.

Supplementary Fig. 4e Time-dependent changes in the hydrodynamic size (red) and polydisperse index (PDI, grey) of the co-assembled supramolecular complex formed by fm-17Abp and fm-SRAL at 4 $^{\circ}$ C

Supplementary Fig. 4d The critical aggregation concentration (CAC) of the co-assembled complex comprising fm-SRAL and fm-17Abp (molar ratio 1:1). Peptides were incubated at 4 $^{\circ}$ C for 12 h and then detected by dynamic light scattering (DLS) for CAC calculation.

4. In Fig3g, the structure of 17A-Lytaca is too small and hard to figure out the structure.

Response. The figure has been modified (new Fig. 3a).

new Fig. 3a

5. In mouse experiments, I cannot find the critical advantage of 17A-Lytaca over MTX. Is the difference in Fig 4c between the two groups significant? Fig 4h and CCL-20 showed better effect of MTX.

Response. We apologize for any confusion caused by the initial presentation of the statistical comparisons in the original Fig. 4c. As the other two reviewers suggested, we have re-conducted the in vivo experiments with an additional control group fm-SRAL. In the revised figures, all meaningful statistical comparisons are now clearly depicted (new Fig. 4).

Regarding the comparison with methotrexate (MTX), a well-established systemic drug for treating psoriasis, it is notable that conventional systemic drug like MTX may exhibit drug-drug interactions and cumulative organ toxicities. In this context, new therapeutics are continually sought to improve safety and efficacy. While the primary focus of this study is on the application of the co-assembly strategy in constructing novel protein degraders, it is noteworthy that 17A-Lytaca demonstrated a comparable therapeutic effect to MTX in psoriasis mice, with the added advantage of better performance in alleviating weight loss (Fig. 4b). Additionally, the ability of 17A-Lytaca to reach and persist in the skin lesion of psoriasis mice (Fig. 4h) underscores its potential tissue selectivity. The versatility of the LYTACA system suggests that further optimization of the co-assembling structures may lead to more tailored therapeutics in our future research.

new Fig. 4

Figure 4. LYTACAs attenuate skin inflammation in psoriasis mouse model via IL-17A clearance. **a**) Experimental timeline. For 7 consecutive days, Saline, fm-17Abp (15 mg/kg/day), fm-SRAL (22 mg/kg/day), 17A-Lytaca (fm-17Abp and fm-SRAL mixture) was given 15 and 22 mg/kg/day after preincubated for 12 h) or MTX (1 mg/kg/twice day) were intraperitoneally administered into BALB/c mice after the topical application of 5% IMQ cream. **b**) Daily body weight changes. **c**) Splenomegaly

degree. **d**) Representative clinical manifestations and H&E staining of the back skin on day 7, scale bar: 100 μ m. **e**) Analysis of acanthosis change on day 7. **f, g**) PASI score reflecting the severity of skin erythema, scaling and thickening for 7 days. **h**) IL-17A protein level in skin lesion on day 7. **i**) Relative mRNA expression of IL-17A, IL-6, CCL-20 and IL-22 in skin lesion. **j, k**) Representative *in vivo* and organ fluorescence images (n = 3) of psoriasis mice at 6 h after intraperitoneal treatment with Cy-labeled 17A-Lytaca, comprising Cy5 labeled fm-17Abp (5 mg/kg) and Cy7 labeled fm-SRAL (7.3 mg/kg). **l, m**) Representative *in vivo* and skin fluorescence images of psoriasis mice or healthy mice at 6 h after intraperitoneal treatment with Cy-labeled 17A-Lytaca, comprising Cy5 labeled fm-17Abp (5 mg/kg) and Cy7 labeled fm-SRAL (7.3 mg/kg). For **b, c** and **e-i**, Data are presented as the mean \pm SD, and error bars represent the standard deviation of biological replicates (n = 5). P values were determined by one-way ANOVA with Tukey's multiple comparisons test (**c, e, h, i**) or two-way ANOVA with Sidak's multiple comparisons test (**b, f, g**). **b** ****P < 0.0001; **c** **P = 0.0013, ****P < 0.0001; **e** ****P < 0.0001; **f** ****P < 0.0001; **g** ****P < 0.0001; **h** **P = 0.0027, ****P < 0.0001; **i** **P = 0.0092, ****P < 0.0001. ns, no significance.

6. In 'Notably, the 17A-Lytaca treatment did not show any obvious toxicity to the organs of mice (Supplementary Fig. 4)', the organ toxicity data is Fig S5, not S4.

Response. Revisions have been made.

7. It would be better to perform PD-L1 degradation study in cancer cells, not RAW264.7.

Response. Given the high expression of SR-A and the consequential impact of PD-L1 on suppressing tumor-specific T cell immunity within macrophages (*J. Biomed. Sci.* **2019**, *26*, 78; *Cancer. Res.* **2019**, *79*, 1493-1506.), we initially conducted a proof-of-concept PD-L1 degradation study in RAW264.7 cells, and the results are highly promising.

Responding to the reviewer's suggestion, we extended our investigation to cancer cells. The outcomes revealed that PD-L1-Lytaca effectively degraded PD-L1 in cancer cell lines expressing SR-A, such as A549 (Fig. 5d, Supplementary Fig. 13c) and HepG2 (Supplementary Fig. 11c, Supplementary Fig. 13b). These findings demonstrate the universality of our strategy across diverse cell lines.

To present the experimental results, we added text to the manuscript (Page 11) stating, "Given that scavenger receptors (SRs) are highly expressed on cancer cells^{13, 40, 41}, we selected A549 and HepG2 cells to ascertain the capability of our LYTACA in degrading PD-L1 on tumor cells. Western blot results revealed successfully degradation in both cell lines (Fig. 5d, Supplementary Fig. 11c), with HepG2 exhibiting superior performance, achieving over 50% degradation." and on page 12 stating, "Remarkably, compared with fm-PD-L1-Lytaca, nf-PD-L1-Lytaca exhibited significantly enhanced

degradation efficiency in macrophages (Fig. 5g) and cancer cells (Supplementary Fig. 13b, c).”

Figure. 5d Dose escalation experiment for PD-L1 degradation in A549 cells treated with 10, 25, 50 and 100 μM fm-PD-L1-Lytaca or 50 μM fm-TPP/fm-SRAL for 12 h. **e)** Chemical structure of N-terminal modification group Fmoc and Nap-FF.

Figure. S11c Western blot of PD-L1 in HepG2 cells treated with 50 μM fm-TPP, fm-SRAL or fm-PD-L1-Lytaca for 12 h.

Figure. S13b Western blot of PD-L1 in HepG2 cells treated with 50 μM nf-TPP, nf-SRAL or nf-PD-L1-Lytaca for 12 h.

Figure. S13c Western blot of PD-L1 in A549 cells treated with 50 μM nf-TPP, nf-SRAL or nf-PD-L1-Lytaca for 12 h. Densitometry was used to calculate protein levels, and data were normalized to the PBS control.

8. In development of new molecules, concentration-dependent cell viability data is essential.

Response. To investigate the concentration-dependent cell viability, we treated RAW264.7 cells with different concentration of 17A-Lytaca (0-125 μM) and the result of luminescence cell viability assay indicated that our peptides have no toxicity to the cells (supplementary Fig. 6). This result was incorporated into the manuscript on page 8 as follows: “Combined with the cell viability assay conducted with varying

concentrations of 17A-Lytaca (Supplementary Fig. S6), these data highlight the safety and potential of our LYTACA strategy for selective cellular targeting.”

Figure. S6 17A-Lytaca treatment shows no obvious toxicity to cells. Cell viability was measured in RAW264.7 cells treated with 0, 25, 50, 75, 100 or 125 μM 17A-Lytaca for 6 h. Data are presented as the mean ± SD, and error bars represent the standard deviation of biological replicates (n = 3). P values were determined by one-way ANOVA with Tukey’s multiple comparisons test. ns, no significance.

REVIEWERS' COMMENTS

Reviewer #1 (Remarks to the Author):

The authors have responded to all my questions. Thanks!

Reviewer #2 (Remarks to the Author):

Response to authors' rebuttal:

The cited papers only speak of linker dependencies in the context of either a highly engineered glycan presentation strategy (biosynthetic installation on N-glycans) or small peptide-ligand conjugates. The majority of extracellular degraders are antibody-based with either protein domains fused or chemical linkers conjugated, neither of which has demonstrated linker issues. The authors should be even more precise about which conjugates exactly have observed a linker-dependency. One can hardly conclude from these specialized systems that a pervasive problem exists. Beyond these text edits, the authors have experimentally addressed all of my comments.

Reviewer #3 (Remarks to the Author):

The authors addressed my questions. However several newly provided figures are for reviewers only. These data should be included and discussed. It is not evident why they are not part of the manuscript. Notably, it is important to include data R1a, b (also asked by reviewer 1) showing that there is no depletion of IL-17 from the medium upon IL-17A-Lytaca treatment – actually, these data show that there is even a substantial increase of IL-17A in the medium upon incubating samples with the very high dose of IL-17A-Lytaca, which is surprising. The results from Figure R5 showing the control Lytaca peptide effect in in vivo models should be included in the manuscript, as it is an important control.

Reviewer #4 (Remarks to the Author):

The manuscript was revised and improved well. All the responses and changes are OK.

Author Responses to Reviewer Comments

Reviewer #2 (Remarks to the Author):

The cited papers only speak of linker dependencies in the context of either a highly engineered glycan presentation strategy (biosynthetic installation on N-glycans) or small peptide-ligand conjugates. The majority of extracellular degraders are antibody-based with either protein domains fuses or chemical linkers conjugated, neither of which has demonstrated linker issues. The authors should be even more precise about which conjugates exactly have observed a linker-dependency. One can hardly conclude from these specialized systems that a pervasive problem exists. Beyond these text edits, the authors have experimentally addressed all of my comments.

Response. We appreciate the reviewer's suggestions. To make a more precise statement about the linker issues in extracellular degraders, we have modified the corresponding text to be more specific: "In the LYTACs strategy, researchers have emphasized the critical role of linker length in the glycan- or peptide-based conjugates for receptor binding and the receptor-mediated degradation of targets" has been added to manuscript on Pages 1.

Reviewer #3 (Remarks to the Author):

The authors addressed my questions. However, several newly provided figures are for reviewers only. These data should be included and discussed. It is not evident why they are not part of the manuscript. Notably, it is important to include data R1a, b (also asked by reviewer 1) showing that there is no depletion of IL-17 from the medium upon IL-17A-Lytaca treatment—actually, these data show that there is even a substantial increase of IL-17A in the medium upon incubating samples with the very high dose of IL-17A-Lytaca, which is surprising. The results from Figure R5 showing the control Lytaca peptide effect in in vivo models should be included in the manuscript, as it is an important control.

Response. We appreciate the reviewer’s suggestion. We have incorporated the suggested figures and enriched the accompanying discussion into the manuscript, as illustrated below:

1. Descriptions about the change of IL-17A and surface SR-A has been added to manuscript on Pages 3 as follows: “Moreover, the process was time-dependent, with the intracellular IL-17A fluorescence signal reaching the maximum at 6 h and then decreased to the initial level at 12 h (Fig. 1d). Contrastingly, the IL-17A level in the culture medium exhibited no discernible alterations, as confirmed by western blot analysis (Supplementary Fig. 2), indicating an overdose of the exogenously added protein. Simultaneously, we monitored changes in SR-A abundance through flow cytometry. The results revealed SR-A recovery at 48 h following a decline at 24 h (Supplementary Fig. 3), supporting the idea of SR-A recycling during the degradation process. Consequently, we propose that the decline in intracellular IL-17A signal is likely associated with the consumption and reduced concentration of our 17A-Lytaca during the latter stage of the time-course experiment, rather than a depletion of SR-A. Notably, in clinical settings, the reported concentration of IL-17A in psoriatic lesions is 0.01 ng/ml,²³ significantly lower than the concentration (0.5 µg/ml) we used in our in vitro cellular assay. Therefore, the efficiency of LYTACA would be sufficient to achieve the degradation of IL-17A at levels relevant to those observed in patients.”

Supplementary Fig. 2 the medium IL-17A exhibited no discernible alterations, suggesting an overdose of the provided protein. a) Medium IL-17A measurement by western blot in the time-

course experiment for IL-17A uptake. RAW264.7 cells incubated with 0.5 $\mu\text{g/ml}$ IL-17A and 50 μM mixed fi-SRAL and fi-17Abp for 1, 3, 6, 12, 24 and 48 h. Medium IL-17A was taken at the indicated time and analyzed by western blot. **b)** The corresponding quantified result of **a)**. Densitometry was used to calculate protein levels, and data ($n = 3$ biologically independent experiments) were normalized to the GAPDH.

Supplementary Fig. 3 fi-SRAL can be taken up and transported to lysosome. Cell surface SR-A receptor quantification via flow cytometry in the timecourse experiment for IL-17A uptake. RAW264.7 cells incubated with 0.5 $\mu\text{g/ml}$ IL-17A and 50 μM mixed fi-SRAL and fi-17Abp for 1, 3, 6, 12, 24 and 48 h. SR-A was measured at the indicated time.

2. Details regarding the co-polymer stoichiometry and its impact on intake efficiency has been added to manuscript on Pages 5 as follows: “Furthermore, we investigated the stoichiometry of the co-assembled complex in varying molar ratios. After a 12-hour incubation, the samples underwent centrifugation to remove the aggregates and subsequent ultrafiltration to eliminate the residual peptides (Supplementary Fig. 6f). High-performance liquid chromatography (HPLC) analysis of the resulting materials revealed a stoichiometry of approximately 1:7 for fm-SRAL and fm-17Abp (Supplementary Fig. 6g). The impact of 17A-Lytaca stoichiometry, with varying molar ratios, on IL-17A intake efficiency was also evaluated, confirming that the optimal ratio for preparing the complex is 1:1 of fm-SRAL and fm-17Abp (Supplementary Fig. 6h). Collectively, these results support the notion that mixed fm-SRAL and fm-17Abp peptides can co-assemble into nanoparticles, which may be the active form that interacts with both IL-17A and SR-A, thus facilitating the cell uptake of extracellular IL-17A.”

Supplementary Fig. 6 The molecular mechanism of mixed 17Abp and SRAL for facilitating the IL-17A endocytosis. a) Uptake of IL-17A in RAW264.7 cells after 6 h of treatment with 0.5 μg/ml IL-17A and 50 μM fi-SRAL and fi-17Abp with different pre-incubation time (2 h, 4 h, 8 h). **b)** Visualization of FRET phenomenon of peptide fi-17Abp and rb-SRAL in RAW264.7 cells by confocal microscopy. Scale bar: 5 μm. **c)** Representative TEM images (n = 3) for mixed fm-SRAL and fm-17Abp in H₂O or sodium dodecylbenzene sulfonate (SDBS, 800 μM) buffer at a concentration of 400 μM. Scale bar, 100 nm. **d)** The critical aggregation concentration (CAC) of

the co-assembled complex comprising fm-SRAL and fm-17Abp (molar ratio 1:1). Peptides were incubated at 4 °C for 12 h and then detected by dynamic light scattering (DLS) for CAC calculation (n = 3). **e**) Time-dependent changes in the hydrodynamic size (red) and polydisperse index (PDI, grey) of the co-assembled supramolecular complex formed by fm-17Abp and fm-SRAL at 4 °C (n = 3). **f**) Schematic illustration of the work flow about stoichiometry determination. ACN, acetonitrile; HPLC, high-performance liquid chromatography. **g**) HPLC analysis of the co-assembled complex with molar ratios of fm-17Abp to fm-SRAL at 1:4 (160 μM: 640 μM), 1:2 (267 μM: 533 μM), 1:1 (400 μM: 400 μM), 2:1 (533 μM: 267 μM), and 4:1 (640 μM: 160 μM). **h**) Ratio effect assay for IL-17A uptake in RAW264.7 cells incubated with 0.5 μg/ml IL-17A and 17A-Lytaca with molar ratios of fm-17Abp to fm-SRAL at 1:4 (160 μM: 640 μM), 1:2 (267 μM: 533 μM), 1:1 (400 μM: 400 μM), 2:1 (533 μM: 267 μM), and 4:1 (640 μM: 160 μM). For **a** and **h**, MFI was determined by flow cytometry and the relative MFI was normalized to the control group. Data are presented as the mean ± SD, and error bars represent the standard deviation of biological replicates (n = 3). P values were determined by one-way ANOVA with Tukey's multiple comparisons test.

3. Descriptions about the covalent chimera linker length screening has been added to manuscript on Pages 6 as follows: “Conversely, the Fmoc modified covalent peptide chimera with varying linker lengths failed to promote the uptake of IL-17A (Supplementary Fig. 8). This observation may be explained by the intrinsic property of SR-A, which requires multivalent binding provided by a branched polyanionic ligand, as seen in the reported polymer-based structure.²⁰ Hence, the single chain peptide in the covalent chimera might not be sufficient to bind SR-A effectively.”

Supplementary Fig. 8 Fmoc modified covalent peptide chimera with different linker length cannot promote the uptake of IL-17A. **a)** Structures of designed covalent chimeras with various conjugate linkers. **b)** Experiment for linker length screening. RAW264.7 cells were exposed to 0.5 µg/ml IL-17A and 50 µM each of peptides C6, C9, and C12. Intracellular IL-17A signal was evaluated using flow cytometry.

4. Descriptions about whether the 17A-Lytaca treatment would leads to cellular activation has been added to manuscript on Pages 6 as follows: “Notably, the treatment of 17A-Lytaca activated the SR-A-mediated signaling response, evidenced by an increase in extracellular regulated protein kinase (ERK) phosphorylation (Supplementary Fig. 7), further supporting the involvement of SR-A in our system.”

Supplementary Fig. 7 17A-Lytaca particles trigger SR-A-mediated activation of signaling responses. **a)** Schematic illustration of the cellular activation of ERK signaling pathway by SR-A-mediated uptake of 17A-Lytaca. **b)** Western blot of ERK and p-ERK in RAW264.7 cells treated with 50 µM 17A-Lytaca, fm-17Abp or fm-SRAL alone for 6 h (n = 3). This finding indicated cellular activation induced by 17A-Lytaca, aligning with previous studies (*Nat. Commun.* **2023**, *14*, 4895; *Respir. Res.* **2008**, *9*, 59) where the ERK signaling pathway was implicated in SR-A-mediated uptake. ERK, extracellular regulated protein kinase; p-ERK, phospho-ERK.

5. Descriptions about the control-Lytaca peptide effect in *in vivo* models has been added to manuscript on Pages 6-7 as follows: “Accordingly, we employed an IMQ-induced psoriasis-like mouse model³⁶, where BALB/c mice were treated with IMQ cream on their shaved back skin for seven consecutive days, followed by intraperitoneal injection of 17A-Lytaca, control-Lytaca (co-assembly prepared from peptide fm-nbp that does not bind to IL-17A, Supplementary Fig. 11), methotrexate (MTX), fm-17Abp alone,

fm-SRAL alone, or saline as control (Figure 4a).”... “Interestingly, treatment with control-Lytaca, fm-17Abp or fm-SRAL alone showed no significant difference compared to the IMQ model group, highlighting the superiority of our co-assembly strategy over inhibitor treatment.” ... “In contrast, the control-Lytaca, fm-SRAL- or fm-17Abp-treated group exhibited no obvious effects on changing IL-17A levels compared to the IMQ model group.” ... “Conversely, the control-Lytaca, fm-17Abp and fm-SRAL alone showed no difference from the positive group (Fig. 4i).”

Supplementary Fig. 11 Design of control-Lytaca with fm-nbp showing no binding to IL-17A.

a) Sequence of fm-nbp. **b)** Representative TEM images for fm-nbp and control-Lytaca particles (mixture of fm-nbp and fm-SRAL) at a concentration of 400 μ M. Scale bar, 100 nm. **c)** SPR curve of fm-nbp at the indicated concentration binding to mouse IL-17A immobilized on the CM5 chip surface. fm-nbp, Fmoc-modified IL-17A no binding peptide.

Figure 4. LYTACAs attenuate skin inflammation in psoriasis mouse model via IL-17A clearance. **a)** Experimental timeline. For 7 consecutive days, Saline, fm-17Abp (15 mg/kg/day), fm-SRAL (22 mg/kg/day), 17A-Lytaca (fm-17Abp and fm-SRAL mixture was given 15 and 22 mg/kg/day after preincubated for 12 h), control-Lytaca (fm-nbp and fm-SRAL mixture was given 12 and 22 mg/kg/day after preincubated for 12 h) or MTX (1 mg/kg/twice day) were intraperitoneally administered into BALB/c mice after the topical application of 5% IMQ cream. **b)**

Daily body weight changes. Data are presented as the mean \pm SEM, and error bars represent the standard deviation of biological replicates (n = 5). **c**) Splenomegaly degree. **d**) Representative clinical manifestations and H&E staining of the back skin on day 7, scale bar: 100 μ m. **e**) Analysis of acanthosis change on day 7. **f, g**) PASI score reflecting the severity of skin erythema, scaling and thickening for 7 days. **h**) IL-17A protein level in skin lesion on day 7. **i**) Relative mRNA expression of IL-17A, IL-6, CCL-20 and IL-22 in skin lesion. **j, k**) Representative *in vivo* and organ fluorescence images (n = 3) of psoriasis mice at 6 h after intraperitoneal treatment with Cy-labeled 17A-Lytaca, comprising Cy5 labeled fm-17Abp (5 mg/kg) and Cy7 labeled fm-SRAL (7.3 mg/kg). **l, m**) Representative *in vivo* and skin fluorescence images of psoriasis mice or healthy mice at 6 h after intraperitoneal treatment with Cy-labeled 17A-Lytaca, comprising Cy5 labeled fm-17Abp (5 mg/kg) and Cy7 labeled fm-SRAL (7.3 mg/kg). For **c** and **e-i**, data are presented as the mean \pm SD, and error bars represent the standard deviation of biological replicates (n = 5). P values were determined by one-way ANOVA with Tukey's multiple comparisons test (**c, e, h, i**) or two-way ANOVA with Sidak's multiple comparisons test (**b, f, g**). Source data are provided as a Source Data file.